# Birth of a Transformer: A Memory Viewpoint

**Alberto Bietti**[*]     **Vivien Cabannes**     **Diane Bouchacourt**     **Hervé Jégou**     **Léon Bottou**
Flatiron Institute          FAIR, Meta              FAIR, Meta              FAIR, Meta          FAIR, Meta

## Abstract

Large language models based on transformers have achieved great empirical successes. However, as they are deployed more widely, there is a growing need to better understand their internal mechanisms in order to make them more reliable. These models appear to store vast amounts of knowledge from their training data, and to adapt quickly to new information provided in their context or prompt. We study how transformers balance these two types of knowledge by considering a synthetic setup where tokens are generated from either global or context-specific bigram distributions. By a careful empirical analysis of the training process on a simplified two-layer transformer, we illustrate the fast learning of global bigrams and the slower development of an "induction head" mechanism for the in-context bigrams. We highlight the role of weight matrices as associative memories, provide theoretical insights on how gradients enable their learning during training, and study the role of data-distributional properties.

## 1   Introduction

As large language models (LLMs) are growing in usage and deployment, it is increasingly important to open the black box and understand how they work. A better understanding can help with interpretability of how these models make decisions, and will be crucial to improve these models and mitigate their failure cases, such as hallucinations or reasoning errors.

An important ingredient in the success of recent LLMs is their ability to learn and reason from information present in their context [6]. These "in-context" learning capabilities are often attributed to the transformer architecture [52], in particular its self-attention blocks, which are able to carefully select parts of the input sequence in order to infer plausible next tokens. Additionally, predictions may require "global" knowledge, such as syntactic rules or general facts, which may not appear in the context and thus needs to be stored in the model.

In order to better understand how transformers develop these capabilities during training, we introduce a synthetic dataset that exhibits both aspects. It consists of sequences generated from a bigram language model, but where some of the bigrams are specific to each sequence. Then, the model needs to rely on in-context learning for good prediction on the sequence-specific bigrams, while the global bigrams can be guessed from global statistics conditioned on the current token. While one-layer transformers fail to reliably predict the in-context bigrams, we find that two-layer transformers succeed by developing an *induction head* mechanism [16, 40], namely a "circuit" of two attention heads that allows the transformer to predict b from a context $[\cdots, \mathsf{a}, \mathsf{b}, \cdots, \mathsf{a}]$, and which appears to be ubiquitous in transformer language models [40, 54].

In order to obtain a fine-grained understanding of how this in-context mechanism emerges during training, we further simplify the two-layer architecture by freezing some of the layers at random initialization, including embeddings and value matrices. This focuses our study on attention and feed-forward mechanisms, while avoiding the difficulty of learning representations, which may

---

[*]Work done while at FAIR, Meta.

37th Conference on Neural Information Processing Systems (NeurIPS 2023).

require complex nonlinear dynamics [15, 33, 45]. This simplification also allows us to introduce a natural model for individual weight matrices as *associative memories*, which store input-output or key-value pairs of embeddings through their outer products. Random high-dimensional embeddings are particularly well-suited to this viewpoint thanks to their near-orthogonality. We provide a detailed empirical study of the training dynamics, by measuring how quickly each weight matrix learns to behave as the desired associative memory, studying how this is affected by data-distributional properties, and investigate the order in which layers are learned: the model first finds the right output associations from the current token and from uniform attention patterns, then the attention heads learn to focus on the correct key-value pairs. We then present theoretical insights on this top-down learning process through population gradient dynamics. Despite its simplicity, our setup already provides useful insights on the internal structure of transformer language models and its evolution throughout training, paving the way for a better understanding of LLMs. We hope that our insights may lead to future research and improvements for LLM practitioners, *e.g.*, for optimization algorithms, data pre-processing and selection, interpretability, fine-tuning, and model editing.

In summary, we make the following contributions:

- We introduce a new synthetic setup to study global vs in-context learning: sequences follow bigram language models, where some bigrams change across sequences and others do not.
- We view the transformer's weight matrices as associative memories that learn to store specific pairs of embeddings, and use this to derive a simplified but more interpretable model for our task.
- We empirically study the training dynamics with careful probing: global bigrams are learned first, then the induction head is formed by learning appropriate memories in a top-down fashion.
- We give theoretical insights on training dynamics, showing how a few top-down gradient steps on the population loss can recover the desired associative memories by finding signal in noisy inputs.

**Related work.** After the success of transformer language models for in-context learning was found [6], several works have studied how in-context learning may arise in various contexts [8, 38, 43, 48, 57]. Multiple recent papers have introduced synthetic tasks in order to better understand and interpret transformers [9, 33, 39, 61]. Several works have attempted to understand internal mechanisms in transformers that are responsible for certain behaviors, an area known as "mechanistic interpretability" [16, 17, 35, 39, 40, 54]. Memory and neural networks have a long history of connections [5, 18, 19, 21, 26, 27, 30, 34, 50, 55, 56]. The associative memories we consider bear similarity to [29, 56], though we use continuous input/outputs. The reader may also be interested in Fast Weight programmers [46, 47]. The use of random vectors for storing memories is related to [23]. Our approach to probing based on memory recall is related to techniques in [13, 17], though motivated differently. [14, 32, 36] study statistical and approximation properties of transformers, highlighting benefits of sparse attention patterns, but do not consider training dynamics. [25, 31, 49, 51] provide theoretical analyses of learning dynamics in transformers and other attention models, but consider different data setups and focus on single-layer architectures, while we focus on two-layer models and take a different viewpoint based on associative memories.

## 2 Background

This section provides background on transformer architectures and induction head mechanisms.

**Transformer architecture.** Transformers [52] operate on sequences of embeddings by alternating self-attention operations and token-wise feed-forward layers. We focus on decoder-only, auto-regressive architectures with a causal attention mask, which are commonly used in large language models trained for next-token prediction [6, 11, 41, 42]. We ignore normalization layers in order to simplify the architecture, since its stability benefits are not as crucial in the small models we consider. Given an input sequence of tokens $z_{1:T} \in [N]^T$ of length $T$, where $N$ is the vocabulary size, the transformer operates as follows:

- **Token embeddings**: each discrete token is mapped to a $d$-dimensional embedding via an embedding map $W_E \in \mathbb{R}^{d \times N}$. We will denote the embeddings of tokens $z_t$ by $x_t := w_E(z_t)$, where $w_E(j)$ is the $j$-th column of $W_E$.
- **Positional embeddings**: the positional embeddings $p_t = w_P(t) \in \mathbb{R}^d$ are added to each token embedding depending on its position in the sequence, leading to the following input embeddings:
$$x_t := x_t + p_t = w_E(z_t) + w_P(t). \tag{1}$$

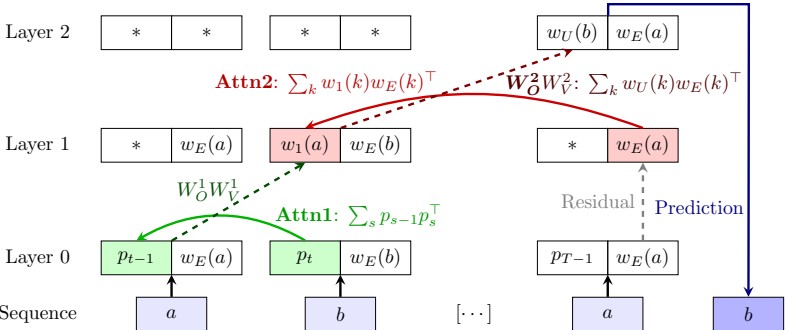

Figure 1: **Induction head mechanism**. Induction heads are a two-layer mechanism that can predict $b$ from a context $[\ldots, a, b, \ldots, a]$. The first layer is a *previous token head*, which attends to the previous token based on positional embeddings ($p_t \to p_{t-1}$) and copies it after a remapping ($w_E(a) \to w_1(a) := W_O^1 W_V^1 w_E(a)$). The second layer is the *induction head*, which attends based on the output of the previous token head ($w_E(a) \to w_1(a)$) and outputs the attended token, remapped to output embeddings ($w_E(b) \to w_U(b)$). Boxes in the diagram represent different embeddings in superposition on each token's residual stream (we omit some irrelevant ones for clarity, *e.g.*, positional embeddings in upper layers), and attention and output associations are shown with the associative memory viewpoint presented in Section 4.

- **Attention blocks**: given an input sequence $x_{1:T} \in \mathbb{R}^{d \times T}$ of embeddings, the causal attention block computes, for $W_K, W_Q, W_V, W_O \in \mathbb{R}^{d \times d}$ (key, query, value, output), and for each $t$,

$$x_t' := W_O W_V x_{1:t} \sigma(x_{1:t}^\top W_K^\top W_Q x_t) \in \mathbb{R}^d, \tag{2}$$

where $\sigma$ takes the softmax of its elements, leading to an attention of the "values" $W_V x_t$ with weights proportional to $\exp((W_K x_s)^\top (W_Q x_t))$. Note that the attention operation usually considers multiple "heads" that each projects the input to a lower dimension. Here we stick to a single head for simplicity, since it will be sufficient for our purposes. Rewriting (2) on each $t$ as $x_{1:T}' = \mathcal{A}(x_{1:T}; W_K, W_Q, W_V, W_O)$, the $\ell$-th layer of the transformer applies attention with layer-specific parameters along with a residual connection as follows:[2]

$$x_{1:T} := x_{1:T} + \mathcal{A}(x_{1:T}; W_K^\ell, W_Q^\ell, W_V^\ell, W_O^\ell)$$

- **Feed-forward blocks**: feed-forward blocks operate on individual token embeddings after each attention block, typically by applying a one-hidden-layer MLP to each token, denoted $\mathcal{F}(\cdot; W_F)$, with a residual connection: at layer $\ell$, we have

$$x_t := x_t + \mathcal{F}(x_t; W_F).$$

Our simplified setup will linear feed-forward layers: $\mathcal{F}(x_t; W_F) = W_F x_t$.

- **Unembedding**: After the last transformer layer, the embeddings are mapped back to the vocabulary space $\mathbb{R}^N$ through a linear "unembedding" layer $W_U = [w_U(1), \ldots, w_U(N)]^\top \in \mathbb{R}^{N \times d}$, where we refer to the $w_U(j)$ as "output embeddings". The output of this layer is then fed into a cross-entropy loss for predicting of $z_{t+1}$ from each $x_t$.

We will sometimes refer to the representations $x_t$ for a given token $t$ throughout layers as its *residual stream* [16], since they consist of sums of embeddings and layer outputs due to residual connections.

**Induction head mechanism.** Induction heads [16, 40] are a particular type of mechanism (or "circuit") in transformers that allows basic in-context prediction of the form $[\cdots, a, b, \cdots, a] \to b$. These were found to be ubiquitous in transformer language models, playing a key role in enabling various forms of in-context learning. The basic mechanism consist of two attention heads in separate layers (see Figure 1 for an illustration): (i) the first is a *previous token head* which attends to the previous token using positional information and copies its embedding to the next token; (ii) the second is the *induction head* itself, which attends using the output of the previous token head, and outputs the original token. Our work focuses on this basic copy mechanism, but we note that richer behaviors are possible, particularly when combining multiple such mechanisms (*e.g.*, [54]).

---

[2]We omit layer indices for simplicity of notation, and use the assignment operator := instead.

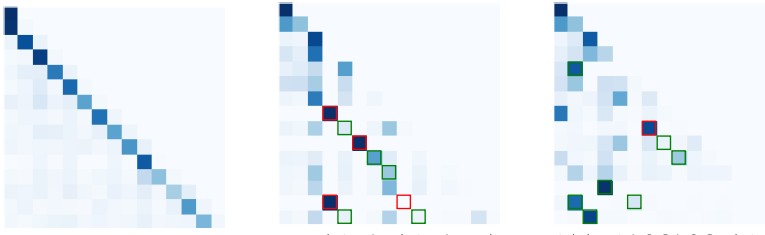

r s a b t s L a b t s L , a b          t h b n t L & C L & C a b t h

Figure 2: **Induction head behavior in attention maps** observed on a 2-layer transformer trained on two variants of our synthetic dataset. Each row shows the attention pattern for predicting the next token. (left) The first layer head always attends to the previous token. (center) For fixed triggers $Q = \{a, t\}$, the second layer head mainly attends to tokens following such triggers. (right) For random triggers, the induction head mechanism is active for any repeated token (here the only trigger is $L$). Red and green boxes highlight tokens following previous occurrences of the query, with red boxes corresponding to "correct" output tokens $o_k$ following trigger tokens $q_k$.

## 3   Synthetic Setup

In this section, we introduce our synthetic data setup, which allows us to carefully study how the induction head mechanism develops during training, and how transformers learn to use information from the context vs simple associations from the training data.

**Bigram data model.**   Our model for sequences consists of a generic bigram language model (*i.e.*, Markov chain), but where the transitions for a few *trigger tokens* denoted $q_k$ are modified in each sequence to always be followed by some *output tokens* $o_k$. Let $K$ be the number of trigger tokens, and fix the following distributions over the vocabulary $[N]$: $\pi_b(\cdot|i)$, $\pi_u$, $\pi_o(\cdot|i)$ and $\pi_q$, for $i \in [N]$. $\pi_b(\cdot|i)$ are the global bigram conditionals, $\pi_u$ the global unigram distribution, while $\pi_o$ is used to sample output tokens at each sequence. The triggers are either fixed to some predefined set of tokens $Q$, or sampled from $\pi_q$. Each sequence $z_{1:T}^n$ is generated as follows:

- (optional) Sample $q_1, \ldots, q_K \sim \pi_q$, i.i.d. without replacement (*random triggers*)
- Sample $o_k \sim \pi_o(\cdot|q_k)$, i.i.d. with replacement.
- Sample $z_1^n \sim \pi_u$ and $z_t^n|z_{t-1}^n \sim p_n(\cdot|z_{t-1}^n)$ for $t = 2, \ldots, T$, where

$$p_n(j|i) = \begin{cases} \pi_b(j|i), & \text{if } i \notin \{q_k\}_k \\ \mathbb{1}\{j = o_k\}, & \text{if } i = q_k. \end{cases}$$

**Experimental setup and initial experiment.**   Our experiments take $\pi_u$ and $\pi_b$ to be unigram and bigram character-level distributions estimated from the tiny Shakespeare dataset, with vocabulary size $N = 65$. We generally sample triggers from $\pi_q = \pi_u$ or fix them to the $K$ most frequent tokens. We sample uniform outputs $o_k$ in most cases, but also experiment with $\pi_o = \pi_b$ in Section 5.

As a preliminary experiment, we train a two-layer vanilla transformer with single-head attention layers and MLP feed-forward layers, following the training setup described in Section 5. On our synthetic data, with fixed (resp. random) triggers and uniform outputs, the model achieves over 99% accuracy (resp. 95%) on output tokens after the first occurrence, versus around 55% for one layer. This gap may be related to the difficulty of modeling three-way interactions with a single attention layer [44]. We visualize attention maps on test sequences in Figure 2, which shows that the model has learned an induction head mechanism. The sequence in the middle figure has $(q_k, o_k) \in \{(a, b), (t, s)\}$. For fixed triggers, the induction head is only active for the triggers used in training, which suggests the presence of a "memory" in the attention layer. For random triggers, it is active on every repeated token, so that the model then needs to disambiguate between in-context and global predictions. For instance, the model may choose to use the retrieved token when it is unlikely to be sampled from the global bigram distribution, something which we found to often be the case in practice.

## 4   The Associative Memory Viewpoint

In this section, we present our associative memory view on transformers: with nearly orthogonal embeddings, the weight matrices behave as associative memories which store pairs of embeddings as

a weighted sum of their outer products. We then introduce a simplified transformer model with fixed random embeddings that will yield a precise understanding of learning dynamics using this viewpoint.

## 4.1 Weight matrices as associative memories

While intermediate representations in the transformer consist of high-dimensional vectors in residual streams, they are often "collapsed" down to scalar measurements by testing against other representations, using operations of the form $v_j^\top W u_i$ for some matrix $W$. For instance, $u_i$ and $v_j$ could be key and query vectors in an attention head, or input and output embeddings for predicting the next token. If $(u_i)_i$ and $(v_j)_j$ are orthonormal (or nearly-orthonormal) sets of embeddings, a natural way to store desired input-output associations $i, j$ is through the following associative memory:

$$W = \sum_{i,j} \alpha_{ij} v_j u_i^\top, \tag{3}$$

so that the scores $v_j^\top W u_i \approx \alpha_{ij}$ may be used to assess the relevance of the $(i, j)$ pair, *e.g.*, as part of a softmax operation in attention or next token prediction.

**Random embeddings.** A simple way to ensure that embeddings $(u_i)_i$ and $(v_j)_j$ are nearly-orthonormal is to set them to be random high-dimensional vectors, such as Gaussian vectors with variance $1/d$ in $d$ dimensions. Indeed, these are known to satisfy [23, 53]

$$u_i^\top u_i \approx 1 \quad \text{and} \quad u_i^\top u_j \approx O\left(\frac{1}{\sqrt{d}}\right),$$

so that (3) is a reasonable way to define an associative memory, without requiring an explicit activation function as employed in end-to-end memory networks [50]. We may also easily create a "remapping" of an existing embedding $u_i$ by multiplying it by a random matrix $W_0 \in \mathbb{R}^{d \times d}$ with Gaussian entries of variance $1/d$, which is commonly used for initializing neural network parameters. The new **remapped embedding** $W_0 u_i$ is near-unit norm, and is near-orthogonal to $u_i$ in addition to the other $u_j$. Note that this fact implies that attention scores at initialization are near-uniform. See Appendix A for more details.

**Learning associative memories.** We now show that learning associations of input-output embeddings via gradient descent leads to a weighted associative memory of a form similar to (3).

**Lemma 1** (Gradients and associative memories). *Let $p$ be a data distribution over input-output tokens, and consider the following loss, where the input and output embeddings $W_E$ and $W_U$ are fixed:*

$$L(W) = \mathbb{E}_{(z,y)\sim p}[\ell(y, W_U W w_E(z))], \tag{4}$$

*with $\ell$ the cross-entropy loss. The gradients of the population loss $L$ then take the form*

$$\nabla_W L(W) = \sum_{k=1}^{N} \mathbb{E}_z[(\hat{p}_W(y = k|z) - p(y = k|z))w_U(k)w_E(z)^\top], \tag{5}$$

*where $\hat{p}_W(y\!=\!k|x) = \sigma(W_U W w_E(z))_k$ are the model's predicted probabilities. Running gradient descent (with or without weight decay) from initialization $W_0$ then leads to estimates of the following form, for some $\alpha_0$ and $\alpha_{ij}$ that vary with the number of iterations:*

$$\hat{W} = \alpha_0 W_0 + \sum_{i,j} \alpha_{ij} w_U(j) w_E(i)^\top. \tag{6}$$

Note that (4) is a convex problem in $W$, thus with appropriate step-size and large enough number of steps (with no weight decay) we can expect gradient descent to be close to the global minimum. At the optimum, if the embeddings are nearly orthogonal, then (5) implies $\hat{p}_W(y = k|z) \approx p(y = k|z)$. We remark that if $W_0$ is a Gaussian random matrix, as if often the case for neural network layers, the first term in (6) plays a minor role: testing $W_0$ against an input-output pair $(i, j)$ with $\alpha_{ij} \neq 0$ will concentrate around zero when $d$ is large, while the $(i, j)$ term in the sum will concentrate around $\alpha_{ij}$. We also note that the gradient updates described above correspond to a so-called maximal feature learning regime similar to $\mu$P updates in intermediate layers of deep networks [58, 59].

**Handling superposition.** In Lemma 1, we assumed that inputs to the matrix $W$ are embeddings of a single token. Yet, in transformer models, the inputs to weight matrices are often sums, or *superpositions* of embeddings. For instance, the initial representations of each token are sums of token and positional embeddings, and representations at later layers are sums of the outputs of each previous block, due to residual connections. Outputs of attention layers are also weighted sums of potentially many embeddings, at least initially when attention patterns are spread out. By linearity, associative memories of the form (6) simply operate individually on each embedding of a superposition, and return a new superposition (up to additional noise due to near-orthogonality). In practice, we will see that learned memories often focus on a single embedding and filter out the rest as noise when irrelevant (see also Section 6). We note that linearity can also be limiting, since it makes it difficult to map sets to specific output embeddings: $u_{\{i,j\}} := u_i + u_j$ needs to map to $Wu_i + Wu_j$, and thus cannot map to a new embedding $v_{\{i,j\}}$. Such mappings of sets thus require non-linear associative memories, for instance by leveraging a sparse decoding of which elements are actually present (*e.g.*, using compressed sensing), or by using MLPs with non-linear activations [15, 30].

## 4.2 A simplified two-layer transformer architecture

We consider a simpler two-layer transformer which is more interpretable with the memory viewpoint, and will help us analyze learning dynamics both empirically and theoretically.

- We freeze input, output and positional embeddings ($W_E, W_U, W_P$) to their random initialization throughout training. This brings us to the Gaussian random vector setup presented above.
- We fix $W_Q^1 = W_Q^2 = I_d$, so that $W_K^1$ and $W_K^2$ play the role of both key and query matrices. This changes the gradient dynamics, but simplifies the model by avoiding the redundancy in (2). The pre-softmax attention scores then take the form $x_q^\top W_K^\ell x_k$, with $x_q$ (resp. $x_k$) the query (resp. key) embeddings, which now directly resembles an associative memory lookup.
- We freeze $W_V^1, W_O^1$, and $W_V^2$ to random initialization. These play the role of *remapping* attended tokens into new tokens, since for random $W$ and large $d$, $Wx$ is nearly orthogonal to $x$ and to any other random embeddings independent of $x$.
- We train $W_O^2$, since the outputs of the induction head need to be mapped back into appropriate output embeddings in order to predict the output tokens $o_k$ correctly.
- We use a single linear feedforward layer after the second attention block, with weight matrix $W_F$. This is plausibly the layer responsible for learning the global bigram distributions.

We remark that while this model freezes some parameters at initialization, it is richer than a "lazy" or neural tangent kernel approximation [10, 22, 24] since the model is still highly non-linear in its parameters and, as we will see, induces rich non-linear learning dynamics.

**Solving the bigram problem with associative memories.** We now show how the above architecture can solve the synthetic bigram problem from Section 3 with well-chosen weight matrices. While this is only a hypothetical model, we show in Section 5 that it is surprisingly faithful to the learned model.

Recall that due to residual connections, the inputs to the weight matrices typically consist of superpositions of various embeddings including token embeddings, positional embeddings, or "remapped" versions thereof. These may be viewed as sets, as illustrated in Figure 1, and associative memories can easily ignore certain elements of the set, *e.g.*, ignore token embeddings by only focusing on positional embeddings. The induction head mechanism can be obtained by setting:

$$W_K^1 = \sum_{t=2}^{T} p_t p_{t-1}^\top, \quad W_K^2 = \sum_{k \in Q} w_E(k)(W_O^1 W_V^1 w_E(k))^\top, \quad W_O^2 = \sum_{k=1}^{N} w_U(k)(W_V^2 w_E(k))^\top,$$

(7)

where $Q$ is the set of triggers when they are fixed, or the support of $\pi_q$ when they are random. In words, the first attention layer matches a token to the previous tokens using positional embeddings. The second layer matches the trigger token to a remapping of itself by $W_O^1 W_V^1$, and the output matches a remapping of the input token by $W_V^2$ to the corresponding output token. We remark that one can easily make the attention patterns more peaked on the correct associations by rescaling $W_K^1$ and $W_K^2$. The global bigram statistics can be encoded in the feed-forward layer as follows:

$$W_F = \sum_{i=1}^{N} \sum_{j=1}^{N} \log \pi_b(j|i) w_U(j) w_E(i)^\top.$$

(8)

The question remains of how the model could trade-off predictions from the induction head and from the feed-forward layer, which are added together due to residual connections. With fixed triggers $Q$, we may simply remove all $i \in Q$ from the summation in (8), so that the model exclusively relies on the attention head for all triggers (indeed, the output of $W_O^2$ is in the span of output embeddings, which are nearly orthogonal to the row space of $W_F$). When the triggers can vary across different sequences, choosing between the induction head and the feed-forward layer is more ambiguous as it depends on context, and $W_F$ may try to learn more complex mappings that also use the outputs of $W_O^2$. In practice, we observe that the model often prefers the induction head, unless its output agrees with one of the top predictions from the global bigram, in which case it tends to prefer those.

**Beyond the simplified architecture.** While our simplified architecture already captures the relevant aspects for the bigram model, it lacks some of the components that appear in standard transformers, such as non-linear MLPs, trained embeddings, layer normalization, and joint learning of a factorization $W_K^\top W_Q$ (potentially with low rank matrices $W_K, W_Q \in \mathbb{R}^{d_h \times d}$ with $d_h < d$ as in multi-head attention), instead of a single matrix $W_K$. In practice, transformers also involve many more layers, as well as multiple heads at each self-attention layer. In Appendix D, we discuss how our memory viewpoint naturally extends to such architectural components, and we illustrate in Appendix E that they empirically lead to similar observations. Nonetheless, we focus on our simpler architecture in the main paper due to simplicity of exposition and better interpretability thanks to a clear identifiability of the role of each matrix, which is lost in models with more heads and layers.

## 5 Empirical Study

In this section, we present our empirical analysis of learning dynamics on the bigram data defined in Section 3, for the simplified architecture defined in Section 4.2. See Appendix E for additional results. Our code is available at `https://github.com/albietz/transformer-birth`.

**Experimental setup.** We train our models using mini-batch SGD with momentum, where each batch consists of 512 fresh sequences of length $T = 256$ sampled from our synthetic model. We use a fixed learning rate and weight decay. Hyperparameters are given in Appendix E. Unless otherwise noted, we use $d = 128$, random triggers with $\pi_q = \pi_u$ and uniform output tokens. The reported accuracies and losses are computed over each fresh batch before it is used for optimization, and are averaged over relevant tokens: "in-context accuracy/loss" numbers only consider predictions of output tokens on triggers starting at the second occurrence (the first is non-deterministic), while "global loss" refers to average loss on non-trigger tokens.

**Memory recall probes.** In addition to loss and accuracy, we consider metrics to check whether individual matrices have learned the desired associative memories: for a desired target memory $W_* = \sum_{(i,j) \in \mathcal{M}} v_j u_i^\top$, the corresponding recall metric is computed from the empirical estimate $\hat{W}$ as

$$R(\hat{W}, W_*) = \frac{1}{|\mathcal{M}|} \sum_{(i,j) \in \mathcal{M}} \mathbb{1}\{\arg\max_{j'} v_{j'}^\top \hat{W} u_i = j\}. \tag{9}$$

We use this for each matrix in (7) as target, and additionally test the previous token matrix $W_K^1$ on smaller time windows. For the final feed-forward layer, we measure the average KL divergence between the predicted softmax distribution using only $W_F$ and the global bigram distribution $\pi_b$:

$$d_{KL}(W_F, \pi_b) := \frac{1}{N} \sum_{k=1}^{N} d_{KL}(\sigma(W_U W_F w_E(k)), \pi_b(\cdot|k)). \tag{10}$$

**Emergence of the induction head via top-down learning.** We begin our study by only training to minimize the loss on *trigger-output* token predictions after their first occurrence. This should be predictable with 100% accuracy using the two-layer induction head mechanism according to Section 4. We also remove the feed-forward layer, in order to focus on the learning of attention matrices $W_K^1$, $W_K^2$ and $W_O^2$ in isolation.

Figure 3 studies the effect of freezing different layers until iteration 300 on the training dynamics. By looking at memory recall probes, we see that training key-query matrices does not lead to any

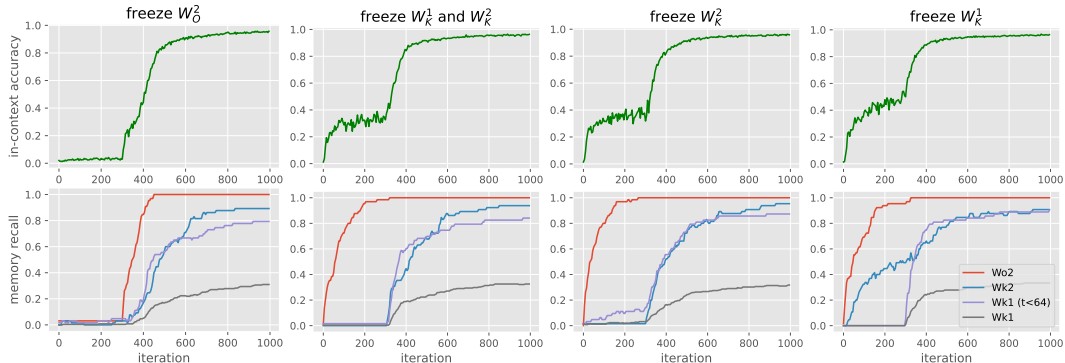

Figure 3: **Learning the induction head alone: in-context accuracy (top) and recall probes (bottom)** with some layers frozen until iteration 300. The output matrix $W_O^2$ can and must be learned before the key-query matrices, but does not suffice for good accuracy. It is easier to learn $W_K^2$ before $W_K^1$, and $W_K^1$ stores initial context positions ($t < 64$) much faster than late positions.

learning unless $W_O^2$ is learned first, and that $W_O^2$ can learn the correct associations even when trained by itself with key-value matrices at random initialization. Recall that the attention weights are essentially uniform when $W_K$ are at random initialization, so that training $W_O^2$ alone resembles a bag-of-words models that aggregates representations throughout the sequence. While such a model has poor prediction accuracy, it is nevertheless sufficient to recover the correct associations in $W_O^2$ (a similar observation was made in [49] in a different setup).

Then, these associations enable learning key-query matrices that focus the attention on relevant tokens, by storing relevant key-query pairs in the form of associative memories, which eventually recovers the desired induction head behavior and leads to near-perfect accuracy. The two rightmost plots suggest that the second layer is learned before the first, in the sense that $W_K^2$ is easier to learn when $W_K^1$ is frozen compared to the reverse, yet learning them together seems beneficial, possibly due to helpful feedback loops [1]. We also observe that $W_K^1$ fits previous token associations for early positions much faster than later positions (purple vs gray line). This is likely due to the fact that it should be enough for the previous token head to attend to the first appearance of each trigger $q_k$, which is typically early in the sequence, so that most of the gradient will focus on early positions.

Overall, this provides a fine-grained understanding of the learning dynamics of induction heads. In Section 6, we analyze how a few gradient steps in a top-down fashion may suffice to recover appropriate associative memories in high dimension and with enough data. See Appendix E for additional experiments, including on the role of dimensionality.

**Global vs in-context learning.** Figure 4(left/right) shows that when training all layers jointly, the global bigram statistics tend to be learned more quickly than the induction head, as seen from the quick drop in loss and KL in early iterations. The $W_O^2$ probe also seems to improve quickly initially, but only leads to mild improvements to in-context predictions. The full learning of the in-context mechanism takes longer, likely due to slower dynamics of the key-query matrices. We also observe a tension between $W_O^2$ and $W_F$ later in training, leading to slight degradations of our probe metrics. This may be due to the fact that the input to $W_F$ now contains additional signal from the induction head which may be leveraged for better predictions, in particular for disambiguation in the case of random triggers, so that our guess of memories in Section 4.2 may no longer be accurate.

**Role of the data distribution.** We can see in Figure 4(left) that changes to the data distribution can have a significant effect on the speed of learning the in-context mechanism. We observe that the following may slow down in-context learning: (i) a smaller number of triggers $K$, (ii) using only rare fixed triggers, and (iii) using random triggers instead of fixed triggers. By inspecting the individual memory probes (see Figure 5 in Appendix E), we hypothesize that (i) and (ii) are due to slow learning of $W_O^2$, while (iii) is more related to slow learning of key-query matrices. This is reasonable since (i-ii) reduce the number of overall output tokens in the data, while (iii) increases the number of possible trigger tokens that should be stored in $W_K^2$, thus increasing the data requirements in order to learn the full associative memory. We also show in Figure 4(center) that changing the output token distribution to bigram distributions at training time reduces the in-context accuracy

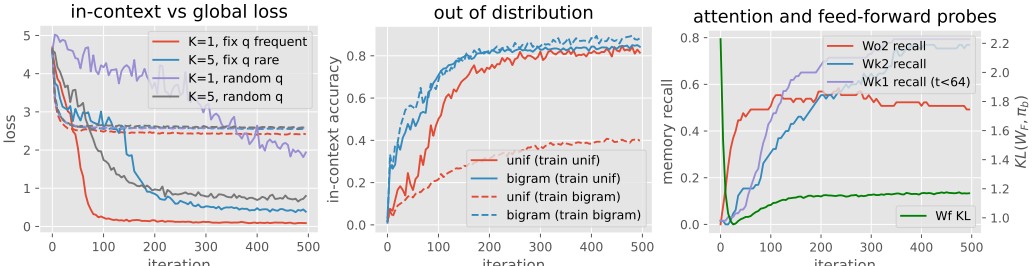

Figure 4: **Global vs in-context learning and data-distributional effects.** (left) Loss on global (dashed) vs in-context (solid) tokens throughout training, for fixed or random trigger tokens $q_k$. The red curves fixes the trigger $q_1$ to the most frequent token, while the fixed triggers in blue curves are less common. (center) In-context accuracy with different training and test distributions $\pi_o$ for output tokens. Uniform leads to better generalization than global bigrams $\pi_b$. (right) Probe metrics throughout training: $W_O^2$ and $W_F$ eventually compete and deviate from our natural estimates.

when using out-of-distribution output tokens, while training on uniform outputs performs well on both distributions. This highlights that using a more diverse training distribution can lead to models with better generalization accuracy, with little additional training cost.

**Additional experiments.** In Appendix E, we provide additional experimental results for varying dimensionality, more complex architectures and training methods, as well as more fine-grained visualizations of the memory associations.

## 6 Theoretical Insights on Learning Dynamics

In this section, we provide theoretical insights on how gradients near initialization may allow the emergence of induction heads, and how this behavior is affected by data-distributional properties.

**Finding signal in noisy inputs.** In Lemma 1, we showed how gradient dynamics on a simple classification task with fixed embeddings of the inputs and outputs lead to associative memories. We now show that when inputs consist of superpositions of multiple embeddings, as is the case in the transformer residual streams, gradients may learn associative memories that filter out irrelevant components of these superpositions, focusing on useful signal instead.

**Lemma 2** (Gradient associative memory with noisy inputs). *Let $p$ be a data distribution on $(x, y) \in \mathbb{R}^d \times [N]$, and consider the following classification problem, with fixed output embeddings $W_U$:*

$$L(W) = \mathbb{E}_{(x,y) \sim p}[\ell(y, W_U W x)].$$

*The gradients take the following form: denoting $\mu_k := \mathbb{E}[x|y=k]$ and $\hat{\mu}_k := \mathbb{E}_x[\frac{\hat{p}_W(k|x)}{p(y=k)}x]$,*

$$\nabla_W L(W) = \sum_{k=1}^{N} p(y=k) w_U(k)(\hat{\mu}_k - \mu_k)^\top.$$

The key takeaway from this lemma is that with enough data (here infinite data), the associative memory arising from gradients can learn to filter out noise from inputs, since it only depends on its expectations or conditional expectations. In particular, $\mu_k$ can isolate relevant parts of $x$ that are predictive of a label $k$, and thus can lead to the right associations.

**An illustrative example.** To gain more intuition about this result, consider the following example: we would like to predict $y$ from $x = w_E(y) + p_t$, where $p_t$ is a positional embedding at a random position $t \in [T]$, which we would like to ignore. Further assume that $y$ is uniformly distributed with $p(y = k) = 1/N$, and consider the matrix obtained after one population gradient step with step-size $\eta$ starting from an initialization $W_0 = 0$ (so that $\hat{p}_{W_0}(k|x) = 1/N$):

$$W_1 = \frac{\eta}{N} \sum_{k=1}^{N} w_U(k)(\mu_k - \bar{\mu})^\top,$$

with $\bar{\mu} = \mathbb{E}[x]$. We show in Appendix B that when $d$ is large enough to ensure near-orthonormal embeddings, we have

$$w_U(k)^\top W_1(w_E(y) + p_t) \approx \frac{\eta}{N} \, \mathbb{1}\{k = y\} + O\left(\frac{1}{N^2}\right),$$

so that for large enough $N$ and $T$, we obtain a near-perfect classifier that ignores the positional embedding, after just one gradient step (but a highly idealized one). Understanding how this translates to the finite dimension and finite sample regime is an important theoretical question that we leave for future work (see [7] for an initial step in that direction). We note that data models related to the above have been useful to study gradient dynamics of neural networks on continuous data [2, 25, 28]. Using a single gradient step to learn representations has also been fruitful in other contexts [3, 12].

**Learning the induction head with gradients.** We may extend the arguments above to show how a few gradient steps can learn the induction head mechanism. We show the following in Appendix B.3.

**Theorem 3** (Learning induction head via three gradient steps, informal). *In a simplified setup, the induction head mechanism as constructed in* (7) *can be learned via sequential gradient steps on the population loss from random initialization, on $W_O^2$, then $W_K^2$, followed by $W_K^1$.*

To show this result, we use Lemma 2 in a similar manner to the illustrative example above to show how training $W_O^2$ by itself at initialization, *i.e.*, when the attention patterns are near-uniform, can recover the desired associative memory. This is possible because when predicting an output token at later occurrences of a trigger, the same output token is guaranteed to be present in the context, while other tokens need not appear more relative to other sequences. See also Figure 9 in Appendix E for numerical experiments verifying this for finite data and dimension. Once $W_O^2$ has learned the correct associations, we show that the gradient with respect to the key-value matrix $W_K^2$ at zero initialization can leverage the correctness of $W_O^2$ to find the right associative memory that focuses the attention on correct triggers. Finally, by linearizing the second-layer attention around $W_K^2 = 0$, we show how gradients w.r.t. $W_K^1$ may learn correct associations for the previous token head.

## 7 Discussion

In this paper, we studied the question of how transformers develop in-context learning abilities, using a simplified setup that allows a fine-grained understanding the model and its training dynamics. While our model already captures rich phenomena at play in the bigram task we consider, more elaborate models are likely needed to understand transformers trained on more complex tasks like language modeling. This includes learning embeddings that are more adapted to the data and more structured (*e.g.*, word embeddings [37, 31], or grokking [33, 39]), factorized key-query and value-output matrices that may induce additional regularization effects [20], and non-linear feedforward layers, which may provide richer associative memories between sets of embeddings. Understanding how transformers leverage such aspects to learn in richer settings is an important next step.

## Acknowledgments and Disclosure of Funding

The authors thank Sainbayar Sukhbaatar and Shubham Toshniwal for helpful discussions.

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

## A Associative Memories with Random Vectors

In this section, we provide basic properties of associative memories based on outer products of random Gaussian embeddings, as described in Section 4.

We consider embeddings $u_k \in \mathbb{R}^d$ with i.i.d. Gaussian $\mathcal{N}(0, \frac{1}{d})$ entries.

We recall a few facts:

- (*Norm*) We have $u_i^\top u_i \approx 1$. This is standard from the concentration of random vectors in high dimension (see, *e.g.*, [53, Theorem 3.1.1]).

- (*Near-orthogonality*) For $i \neq j$, we have $u_i^\top u_j = O(1/\sqrt{d})$. To see this, denoting $u_i = d^{-1/2}(\tilde{u}_{ik})_k$, where $\tilde{u}_{ik}$ are the unnormalized entries of $u_i$, note that we have

$$\sqrt{d} u_i^\top u_j = \frac{1}{\sqrt{d}} \sum_{k=1}^d \tilde{u}_{ik} \tilde{u}_{jk} \to \mathcal{N}(0, 1),$$

  by the central limit theorem, since for each $k$, the quantities $\tilde{u}_{ik} \tilde{u}_{jk}$ are zero-mean, unit-variance, i.i.d. random variables.

- (*Remapping: norm*) If $W$ is a Gaussian random matrix with i.i.d. $\mathcal{N}(0, \frac{1}{d})$ entries, then for any fixed $x$ we have $\|Wx\| \approx \|x\|$. This follows from Johnson-Lindenstrauss (see, *e.g.*, [53, Lemma 5.3.2 and Exercise 5.3.3]). In particular, if $x$ is a normalized Gaussian embedding as above, then $\|Wx\| \approx 1$.

- (*Remapping: near-orthogonality*) Consider a random vector $x = \frac{1}{\sqrt{d}} \tilde{x}$ and a random matrix $W = \frac{1}{\sqrt{d}} \tilde{W}$, where the entries of $\tilde{x}$ and $\tilde{W}$ are i.i.d. $\mathcal{N}(0, 1)$. Then $x$ and $Wx$ are nearly orthogonal. To see this, note that $\mathbb{E}[x^\top W x] = \mathbb{E}[x^\top \mathbb{E}[W] x] = 0$, and the variance is

$$\mathbb{E}(x^\top W x)^2 = \mathbb{E} \sum_{i,j} x_i^2 W_{ij}^2 x_j^2 = \frac{1}{d^3} \mathbb{E} \sum_i \tilde{x}_i^4 \tilde{W}_{ii}^2 + \mathbb{E} \sum_{i \neq j} \tilde{x}_i^2 \tilde{x}_j^2 \tilde{W}_{ij}^2$$

$$= \frac{1}{d^3} \left( d M_4 M_2 + \frac{d(d-1)}{2} M_2^3 \right) = O\left(\frac{1}{d}\right),$$

  where $M_2$ and $M_4$ denote the second and fourth moments of the standard Gaussian, respectively. Then, Chebyshev's inequality implies that $|x^\top W x| = O(1/\sqrt{d})$ with high probability.

Ensuring appropriate memory lookups then requires such properties to hold for many embeddings and pairs of embeddings, with errors that are small enough to ensure correct associations. This may be achieved with careful union bounds or more powerful concentration results. We do not attempt to do this in a precise manner in this paper, and will generally assume $d$ large enough to satisfy the desired associative memory behaviors, noting that a precise analysis is an important direction for future work (see [7] for a first step in this direction).

## B Theoretical Insights on Gradient Dynamics

In this section, we provide additional details on the theoretical insights from Section 6, including details on the illustrative example (Section B.1), derivations of gradients w.r.t. key-query matrices at initialization (Section B.2), as well as a study of how the induction head mechanism may develop in a simplified setup, using a sequence of single layer-wise gradient steps in a top-down manner (Section B.3).

### B.1 Details on illustrative example

Consider the example discussed in Section 6: we would like to predict $y$ from $x = w_E(y) + p_t$, where $p_t$ is a positional embedding at a random position $t \in [T]$, which we would like to ignore. Further assume that $y$ is uniformly distributed ($p(y = k) = 1/N$) and consider the matrix obtained after one population gradient step with step-size $\eta$ starting from an initialization $W_0 = 0$ (so

that $\hat{p}_{W_0}(k|x) = 1/N$):

$$W_1 = \frac{\eta}{N} \sum_{k=1}^{N} w_U(k)(\mu_k - \bar{\mu})^\top, \tag{11}$$

with $\bar{\mu} = \mathbb{E}[x]$.

Note that we have $\mu_k = w_E(k) + \frac{1}{T}\sum_t p_t$ and $\bar{\mu} = \frac{1}{N}\sum_k w_E(k) + \frac{1}{T}\sum_t p_t$, so that (11) becomes

$$W_1 = \frac{\eta}{N} \sum_{k=1}^{N} w_U(k)(w_E(k) - \bar{w}_E)^\top, \tag{12}$$

with $\bar{w}_E := \frac{1}{N}\sum_{k=1}^{N} w_E(k)$. When $d$ is large enough to ensure near-orthonormal embeddings, we have for any $y$ and $t$,

$$W_1(w_E(y) + p_t) \approx \frac{\eta}{N} w_U(y) + O\left(\frac{1}{N^2}\right).$$

This implies

$$w_U(k)^\top W_1(w_E(y) + p_t) \approx \frac{\eta}{N}\, \mathbb{1}\{k = y\} + O\left(\frac{1}{N^2}\right),$$

as claimed in the main text. The classifier $\hat{y} = \arg\max_k w_U(k)^\top W_1(w_E(y)+p_t)$ then has essentially perfect accuracy, and has learned to ignore the spurious positional embeddings, which are simply exogenous noise.

## B.2   Gradients on key-query matrices at initialization

We now derive expressions for population gradients of the attention key-query matrices at zero initialization, noting that random initialization behaves similarly to zero initialization. Although the optimization problems involving these matrices are non-convex, these gradients at initialization lead to associative memories, similar to Lemma 2. When output matrices of the previous layer already encode the desired associations, these gradients can lead to associative memories that focus the attention on the correct key-value pairs.

We begin with the following lemma, which gives the gradient of the loss w.r.t. $W = W_K^2$ at zero initialization. For simplicity, we drop the $d^{-1/2}$ factor from the softmax, which only changes gradients by a multiplicative factor, and thus does not change its form.

**Lemma 4** (Gradient of second attention layer). *Consider the following loss for predicting the next token $y$ from an attention layer with inputs $X = [x_1, \ldots, x_T]$, and value-output matrix $\Phi_2 := W_O^2 W_V^2$:*

$$L(W) = \mathbb{E}_{(X,y)}[\ell(y, \xi(X))], \qquad \xi(X) = W_U \Phi_2 X \sigma(X^\top W x_T), \tag{13}$$

*with $\ell$ the cross-entropy loss and $\sigma(u)_t = \frac{e^{u_t}}{\sum_s e^{u_s}}$ for $u \in \mathbb{R}^T$ is the softmax.*

*The gradient at $W = 0$ is given by*

$$\nabla_W L(W)\big|_{W=0} = \sum_{k=1}^{N} \mathbb{E}_{(X,y)}\left[(\hat{p}_W(k|X) - \mathbb{1}\{y=k\})\frac{1}{T}\sum_{t=1}^{T} w_U(k)^\top \Phi_2 x_t \cdot (x_t - \bar{x}_{1:T})x_T^\top\right]$$

$$= \frac{1}{T}\sum_{k=1}^{N}\sum_{t=1}^{T} \mathbb{E}_X[\hat{p}_W(k|X)w_U(k)^\top \Phi_2 x_t \cdot (x_t - \bar{x}_{1:T})x_T^\top]$$

$$- \frac{1}{T}\sum_{k=1}^{N}\sum_{t=1}^{T} p(y=k)\, \mathbb{E}_X[w_U(k)^\top \Phi_2 x_t \cdot (x_t - \bar{x}_{1:T})x_T^\top \mid y=k]$$

*with $\bar{x}_{1:T} = \frac{1}{T}\sum_{t=1}^{T} x_t$.*

Now we consider the gradient w.r.t. $W = W_K^1$ at zero initialization, and consier a simplification of the second layer attention to its linearization around $W_K^2 = 0$. We will see that this still provides first-order information that is sufficient for $W_K^1$ to be learned.

**Lemma 5** (Gradient of first attention layer). *Consider the following loss for predicting the next token $y$ from a stack of two attention layers, with all parameters fixed except for $W = W_K^1$, the key-query matrix at the first attention layer:*

$$L(W) = \mathbb{E}_{(X,y)}[\ell(y, \xi(X))], \qquad \xi(X) = W_U \Phi_2 X \bar{\sigma}(Z(W)^\top W_2 x_T). \tag{14}$$

*Here, $\bar{\sigma}(u_{1:T})_t = \frac{1}{T}(1 + u_t - \frac{1}{T}\sum_{s=1}^T u_s)$ is the linearization of the softmax around $0$, and $Z(W) = [z_1(W), \ldots, z_T(W)]$ with*

$$z_t(W) = \sum_{s=1}^t \Phi_1 x_s \sigma(p_{1:t}^\top W p_t)_s,$$

*and $\Phi_\ell = W_O^\ell W_V^\ell$ for $\ell = 1, 2$.*

*The gradient at $W = 0$ is given by*

$$
\begin{aligned}
&\nabla_W L(W)\big|_{W=0} \\
&= \sum_{k=1}^N \mathbb{E}_X \left[ \hat{p}_W(k|X) \frac{1}{T} \sum_{t=1}^T w_U(k)^\top \Phi_2 x_t \cdot \frac{1}{t} \sum_{s=1}^t (\Phi_1 x_s)^\top W_2 x_T (p_s - \bar{p}_{1:t}) p_t^\top \right] \\
&\quad - \sum_{k=1}^N p(y = k) \mathbb{E}_X \left[ \frac{1}{T} \sum_{t=1}^T w_U(k)^\top \Phi_2 x_t \cdot \frac{1}{t} \sum_{s=1}^t (\Phi_1 x_s)^\top W_2 x_T (p_s - \bar{p}_{1:t}) p_t^\top \Big| y = k \right] \\
&\quad - \sum_{k=1}^N \mathbb{E}_X \left[ \hat{p}_W(k|X) w_U(k)^\top \Phi_2 \bar{x}_{1:T} \cdot \frac{1}{T} \sum_{t=1}^T \frac{1}{t} \sum_{s=1}^t (\Phi_1 x_s)^\top W_2 x_T (p_s - \bar{p}_{1:t}) p_t^\top \right] \\
&\quad + \sum_{k=1}^N p(y = k) \mathbb{E}_X \left[ w_U(k)^\top \Phi_2 \bar{x}_{1:T} \cdot \frac{1}{T} \sum_{t=1}^T \frac{1}{t} \sum_{s=1}^t (\Phi_1 x_s)^\top W_2 x_T (p_s - \bar{p}_{1:t}) p_t^\top \Big| y = k \right]
\end{aligned}
$$

### B.3 Learning the induction head mechanism

In this section, we analyze the training dynamics of the induction head mechanism, in the following simplified setup: we consider a single trigger ($K = 1$), and assume that $\pi_u$, $\pi_q$, $\pi_o$ and $\pi_b(\cdot|i)$ are uniform over $[N]$ for any $i$.

To further simplify the analysis, we consider a loss that only considers sequences of length $T$ where the last input $z_T$ is the *second occurrence* of the trigger token, and the label $y = z_{T+1}$ is the corresponding output token. We note that this may be easily extended to later occurrencies of the trigger. This is similar to the setup of Figure 1, where the loss is only taken on triggers after the second occurrence: in that case, the loss may be written as a weighted sum of the one we consider here, weighted by the probability of the second (or later) trigger appearing at the given position $T$.

In practice, when the loss is on all tokens and $W_F$ is also learned, we may expect that $W_F$ quickly learns the global bigram statistics, as we saw empirically in Section 5. Indeed, the current token embedding, which is included in the input superposition, has strong predictive signal compared to the attention layers, which initially mainly appear as noise. This is then similar to the setup of Lemma 1, which provides recovery of bigram statistics when $d$ is large (though we note that the other information from attention layers in the inputs may eventually be used and bias away from perfect recovery, see Figure 4(right)). Once such global estimates are obtained, the expected loss will be mainly dominated by trigger tokens, leading to the setup above.

For simplicity, we thus drop the feed-forward layer $W_F$ in the remainder of this section, focusing on the learning of $W_O^2$, $W_K^2$ and $W_K^1$, in this top-down order. We will consider zero-initialization a single gradient steps, noting that random initialization should lead to similar associative memory behaviors when the dimension is large enough, since it leads to a remapping of input embeddings which is near-orthogonal to any output embedding (see Appendix A).

### B.3.1 Learning $W_O^2$

We begin by studying the learning of the second output matrix $W_O^2$. In the above data model, we may consider a loss as in Lemma 2 with input-outputs $(x, y)$, where $y$ is the output token of the sequence,

and $x$ depends on the random sequence $z_{1:T}$ as

$$x = \frac{1}{T} \sum_{t=1}^{T} W_V^2(w_E(z_t) + \varepsilon_t),$$

where $\varepsilon_t = p_t + \frac{1}{t} \sum_{s=1}^{t} \Phi_1(w_E(z_s) + p_s)$ with $\Phi_1 = W_O^1 W_V^1$, is a "noise" vector from the residual streams, containing positional embeddings as well as an average attention output from the first layer. In practice, the logit predictions are of the form $W_U(W_O^2 x + \varepsilon_T)$ due to residual connections, but we ignore the term $W_U \varepsilon_T$ for simplicity, noting that it is near-zero when $d$ is large.

After a gradient step on $W_O^2$ with step-size $\eta$, starting from zero-initialization (so that $\hat{p}(k|x) = p(y = k) = 1/N$ for all $x$), Lemma 2 yields

$$W_O^2 = \frac{\eta}{N} \sum_{k=1}^{N} w_U(k)(\mathbb{E}[x|y=k] - \mathbb{E}[x])^\top. \tag{15}$$

Now, consider the random variables $q$ (trigger token), $o$ (output token), $t_o$ (position of the first occurrence of the output token). In our simplified data model, $q$ and $t_o$ have the same distribution regardless of the conditioning on $y = k$, while $o$ is equal to $k$ when $y = k$, while it is uniform in $[N]$ without this condition. The sequence $z_{1:T}$ has the same distribution in either $p(\cdot)$ or $p(\cdot|y = k)$, except for the token $z_{t_o}$.

Then, we may write:

$$\mathbb{E}[x|y=k] - \mathbb{E}[x] = \frac{1}{T}\left(\mathbb{E}[W_V^2 w_E(z_{t_o})|y=k] - \mathbb{E}[W_V^2 w_E(z_{t_o})]\right)$$

$$+ \frac{1}{T}\left(\mathbb{E}\left[\sum_{t=t_o}^{T} W_V^2 \varepsilon_t | y=k\right] - \mathbb{E}\left[\sum_{t=t_o}^{T} W_V^2 \varepsilon_t\right]\right),$$

since $\varepsilon_t$ is independent of $o$ when $t < t_o$. Noting that $\varepsilon_t$ only depends on $z_{t_o}$ and thus on $y$ through the first layer attention, we have

$$\mathbb{E}[x|y=k] - \mathbb{E}[x] = \frac{1}{T} W_V^2(w_E(k) - \bar{w}_E)$$

$$+ \frac{1}{T}\left(\mathbb{E}\left[\sum_{t=t_o}^{T} \frac{1}{t} W_V^2 \Phi_1 w_E(z_{t_o}) | y=k\right] - \mathbb{E}\left[\sum_{t=t_o}^{T} \frac{1}{t} W_V^2 \Phi_1 w_E(z_{t_o})\right]\right)$$

$$= \frac{1}{T} W_V^2(w_E(k) - \bar{w}_E) + \frac{\tau}{T} W_V^2 \Phi_1(w_E(k) - \bar{w}_E),$$

where $\tau := \mathbb{E}\left[\sum_{t=t_o}^{T} \frac{1}{t}\right]$, and $\bar{w}_E = \frac{1}{N} \sum_{k=1}^{N} w_E(k)$. Thus, (15) becomes

$$W_O^2 = \frac{\eta}{NT} \sum_{k=1}^{N} w_U(k)(W_V^2(w_E(k) - \bar{w}_E))^\top + \frac{\eta\tau}{NT} \sum_{k=1}^{N} w_U(k)(W_V^2 \Phi_1(w_E(k) - \bar{w}_E))^\top, \tag{16}$$

so that when $d$ is learn enough to ensure near-orthonormal embeddings, we have

$$w_U(k)^\top W_O^2 W_V^2 w_E(j) \approx \frac{\eta}{NT} \mathbb{1}\{k=j\} + O\left(\frac{\eta}{N^2 T}\right)$$

$$w_U(k)^\top W_O^2 W_V^2 \Phi_1 w_E(j) \approx \frac{\eta\tau}{NT} \mathbb{1}\{k=j\} + O\left(\frac{\eta\tau}{N^2 T}\right),$$

where the $O(\cdot)$ terms are due to the $\bar{w}_E$ elements. The first line yields a behavior that matches desired associative memory in (7) of Section 4.2 when $N$ is large. The second line shows additional spurious associations that are stored in $W_O^2$ due to the output of the first layer attention, but which may be "cleaned up" once the attention layers start focusing on the correct tokens.

Finally, we note that despite the recovery of these useful associations after one gradient step, the predictions with this estimate $W_O^2$ are still near-random, since in the bag-of-words setup with average attention, the output token cannot be distinguished from any other token in the sequence in our model (except perhaps the trigger token, which is guaranteed to appear twice, but does not provide any signal to infer the output token, since the two are independent).

## B.3.2 Learning $W_K^2$

Now assume that $W_O^2$ is as in (16). As argued above, the predictions $\hat{p}(k|x)$ are essentially random $1/N$ in our model for $W_K^2 = 0$, so that after one gradient step on $W_K^2$ with learning rate $\eta$, Lemma 4 yields:

$$W_K^2 = \frac{\eta}{TN} \sum_{k,t} \left( \mathbb{E}[w_U(k)^\top \Phi_2 x_t \cdot (x_t - \bar{x}) x_T^\top \mid y = k] - \mathbb{E}[w_U(k)^\top \Phi_2 x_t \cdot (x_t - \bar{x}) x_T^\top] \right), \quad (17)$$

where $x_t$ are the inputs to the second attention layer, given by

$$x_t = x_{t,0} + x_{t,1} \tag{18}$$
$$x_{t,0} = w_E(z_t) + p_t \tag{19}$$
$$x_{t,1} = \frac{1}{t} \sum_{s=1}^{t} \Phi_1(w_E(z_s) + p_s). \tag{20}$$

From now on, we consider a simplified architecture where only $x_{t,0}$ are fed as queries and values, while only $x_{t,1}$ are fed as keys. Using the fact that trigger tokens $q$ are sampled uniformly (*i.e.*, $\pi_q = 1/N$), we have

$$W_K^2 = \frac{\eta}{TN} \sum_{k=1}^{N} \sum_{t=1}^{T} \left( \mathbb{E}[A_{t,k} \mid y = k] - \mathbb{E}_X[A_{t,k}] \right) \tag{21}$$

$$= \frac{\eta}{TN^2} \sum_{k=1}^{N} \sum_{t=1}^{T} \sum_{j=1}^{N} \left( \mathbb{E}[A_{t,k} \mid y = k, q = j] - \mathbb{E}[A_{t,k} \mid q = j] \right) \tag{22}$$

where

$$A_{t,k} = w_U(k)^\top \Phi_2 x_{t,0} \cdot (x_{t,1} - \bar{x}_1) x_{T,0}^\top, \tag{23}$$

with $\bar{x}_1 = \frac{1}{T} \sum_t x_{t,1}$. Now, note that we have $w_U(k)^\top \Phi_2 x_{t,0} \approx \alpha \mathbb{1}\{z_t = k\}$ with $\alpha = \eta/TN$ by (16), and $x_{T,0} = w_E(q) + p_T$. This yields

$$W_K^2 \approx \frac{\alpha \eta}{TN^2} \sum_{j=1}^{N} \sum_{k=1}^{N} \Delta_{k,j} (w_E(j) + p_T)^\top, \tag{24}$$

with

$$\Delta_{k,j} := \mathbb{E}\left[ \sum_{t=1}^{T} \mathbb{1}\{z_t = k\}(x_{t,1} - \bar{x}_1) | y = k, q = j \right] - \mathbb{E}\left[ \sum_{t=1}^{T} \mathbb{1}\{z_t = k\}(x_{t,1} - \bar{x}_1) | q = j \right]$$
$$= \Delta_{k,j}^o + \Delta_{k,j}^q + \Delta_{k,j}^r,$$

where the three terms split the sum inside the expectation

$$\Delta_{k,j}^o := \mathbb{E}\left[ \mathbb{1}\{z_{t_o} = k\}(x_{t_o,1} - \bar{x}_1) | y = k, q = j \right] - \mathbb{E}\left[ \mathbb{1}\{z_{t_o} = k\}(x_{t_o,1} - \bar{x}_1) | q = j \right]$$

$$\Delta_{k,j}^q := \mathbb{E}\left[ \sum_{t \in \mathcal{T}_q} \mathbb{1}\{z_t = k\}(x_{t,1} - \bar{x}_1) | y = k, q = j \right] - \mathbb{E}\left[ \sum_{t \in \mathcal{T}_q} \mathbb{1}\{z_t = k\}(x_{t,1} - \bar{x}_1) | q = j \right]$$

$$\Delta_{k,j}^r := \mathbb{E}\left[ \sum_{t \in \mathcal{T}_r} \mathbb{1}\{z_t = k\}(x_{t,1} - \bar{x}_1) | y = k, q = j \right] - \mathbb{E}\left[ \sum_{t \in \mathcal{T}_r} \mathbb{1}\{z_t = k\}(x_{t,1} - \bar{x}_1) | q = j \right],$$

where $\mathcal{T}_q = \{t_o - 1, T\}$ and $\mathcal{T}_r = [T] \setminus \{t_o, t_o - 1, T\}$ (recall that $t_o$ is a random variable, corresponding to the first occurrence of the output token, so that these sets are random).

We will now show that $\Delta_{k,j}^o$ carries the desired signal for the appropriate induction head associative memory, while $\Delta_{k,j}^q$ and $\Delta_{k,j}^r$ are negligible, for $N$ large enough.

**Controlling $\Delta_{k,j}^o$.** For $t = t_o$, noting that $z_{t_o} = y$, we have

$$\Delta_{k,j}^o = \mathbb{E}\left[\mathbb{1}\{y = k\}(x_{t_o,1} - \bar{x}_1)|y = k, q = j\right] - \mathbb{E}\left[\mathbb{1}\{y = k\}(x_{t_o,1} - \bar{x}_1)|q = j\right]$$

$$= \left(1 - \frac{1}{N}\right)\mathbb{E}\left[x_{t_o,1} - \bar{x}_1|y = k, q = j\right]$$

$$= \frac{N-1}{N}\left(\bar{p} + \sum_{i=1}^{N} a_{k,j,i}\Phi_1 w_E(i)\right),$$

with $a_{k,j,i} \approx (\Phi_1 w_E(i))^\top \mathbb{E}\left[x_{t_o,1} - \bar{x}_1|y = k, q = j\right]$ thanks to near-orthonormality, and

$$\bar{p} = \mathbb{E}_{t_o}\left[\frac{1}{t_o}\sum_{s=1}^{t_o} p_s - \frac{1}{T}\sum_{t=1}^{T}\frac{1}{t}\sum_{s=1}^{t} p_s\right]$$

is a spurious positional mixture.

We then distinguish the following cases:

- If $j \neq k$ and $i = j$, since the trigger token $j$ only appears at positions $t_o - 1$ and $T$, we have

$$a_{k,j,i} \approx \mathbb{E}_{t_o}\left[\frac{1}{t_o} - \frac{1}{T}\sum_{t=t_o-1}^{T}\frac{1}{t} - \frac{1}{T^2}\right] =: \gamma_T.$$

  We may expect $t_o$ to be concentrated around $T/2$, in which case $\gamma_T \gtrsim \frac{2}{T} - \frac{1}{T} - \frac{1}{T^2} \geq \frac{C}{T} > 0$ for $T$ larger than a small constant.

- If $j = k = i$, the two occurrences of the trigger happen one after the other, so it must be that $t_o = T$. Then

$$a_{k,j,i} \approx \frac{2}{T} - \frac{1}{T(T-1)} - \frac{2}{T^2} = \frac{2}{T} + O\left(\frac{1}{T^2}\right),$$

  for $T$ larger than a small constant.

- If $i \neq j = k$, all tokens up to position $t_o - 2 = T - 2$ are i.i.d. uniform in $[N] \setminus \{j\}$, so that

$$a_{k,j,i} \approx \frac{T-2}{T(N-1)} - \frac{1}{T}\left((T-2)\cdot\frac{1}{N-1} + \frac{T-2}{(T-1)(N-1)} + \frac{T-2}{T(N-1)}\right) = O\left(\frac{1}{N}\right)$$

- If $i \neq j$ and $j \neq k$, all tokens except at positions $t_o - 1, t_o$ and $T$ (we have $t_o < T$) are uniform in $[N] \setminus \{j\}$. The triggers do not contribute anything to $a_{k,j,i}$ since $i \neq j$, and the output token may be also randomized by taking the average over $k \in [N] \setminus \{j\}$. We thus obtain

$$\frac{1}{N-1}\sum_{k \neq j} a_{k,j,i} \approx O\left(\frac{1}{N}\right).$$

In summary, we obtain

$$\frac{1}{N}\sum_{k=1}^{N} a_{k,j,i} \approx \begin{cases} O\left(\frac{1}{N}\right), & \text{if } i \neq j \\ \Omega\left(\frac{1}{T}\right), & \text{if } i = j. \end{cases}$$

Thus, when $N$ is large, while $T$ is moderate, the above sum leads to more signal in the $i = j$ terms compared to $i \neq j$. In particular, this yields

$$(\Phi_1 w_E(i))^\top\left(\frac{1}{N}\sum_{k=1}^{N}\Delta_{k,j}^o\right) \approx \begin{cases} O\left(\frac{1}{N}\right), & \text{if } i \neq j \\ \Omega\left(\frac{1}{T}\right), & \text{if } i = j, \end{cases}$$

so that this component in (24) acts precisely like the desired associative memory in (7).

It remains to show that the other components are negligible compared to this. It then suffices to show:

$$\frac{1}{N}\sum_{k=1}^{N}(\Delta_{k,j}^q + \Delta_{k,j}^r) \approx o\left(\frac{1}{T}\right).$$

**Controlling $\Delta^q_{k,j}$.** For $t \in \mathcal{T}_q$, note that we always have $z_t = j$ in the expectations, so that $\Delta^q_{k,j} = 0$ unless $k = j$. For $k = j$, we have $\Delta^q_{k,j} = O(1)$, so that

$$\frac{1}{N} \sum_{k=1}^N \Delta^q_{k,j} = O\left(\frac{1}{N}\right).$$

**Controlling $\Delta^r_{k,j}$.** Using that $\|x_{t,1} - \bar{x}_1\| \leq C = O(1)$ for all $t$, we provide the following crude bound via the triangle inequality and Hölder inequality:

$$\Delta^r_{k,j} = \mathbb{E}\left[\sum_{t \in \mathcal{T}_r} \mathbb{1}\{z_t = k\}(x_{t,1} - \bar{x}_1)|y = k, q = j\right] - \mathbb{E}\left[\sum_{t \in \mathcal{T}_r} \mathbb{1}\{z_t = k\}(x_{t,1} - \bar{x}_1)|q = j\right]$$

$$\|\Delta^r_{k,j}\| \leq C\left(\mathbb{E}\left[\sum_{t \in \mathcal{T}_r} \mathbb{1}\{z_t = k\}|y = k, q = j\right] + \mathbb{E}\left[\sum_{t \in \mathcal{T}_r} \mathbb{1}\{z_t = k\}|q = j\right]\right) \leq \frac{2CT}{N},$$

since $z_t$ is independent of $y$ given $t \in \mathcal{T}_r$ and thus is uniform in $[N] \setminus \{j\}$, and $|\mathcal{T}_r| \leq T$. We note, however, that $\Delta^r_{k,j}$ may be controlled much more finely by leveraging the similarities between the distributions of $z_t, t \in \mathcal{T}_r$ with or without conditioning on $y$.

Overall, we have shown that up to some spurious positional embeddings, $W_K^2$ behaves as the desired associative memory from (7) when $N$ is large enough, satisfying:

$$(\Phi_1 w_E(i))^\top W_K^2 w_E(j) \approx \frac{\alpha\eta}{TN}\left\{\Omega\left(\frac{1}{T}\right)\mathbb{1}\{i = j\} + O\left(\frac{T}{N}\right)\right\} \tag{25}$$

We note that one may then amplify the gap between correct and incorrect associations by having a large enough step-size, which then makes the softmax more peaked and hence the attention more sparse and focused on correct associations.

### B.3.3 Learning $W_K^1$

We now assume that $W_O^2$ and $W_K^2$ have learned the correct associations, and consider one gradient step away from zero-initialization on $W_K^1$. Note that when $W_K^1 = 0$, the predictions of the model are still often near random chance. Indeed, the second layer attention will attend to all tokens starting at the first occurrence of the trigger, since all such tokens contain $\Phi_1 w_E(q)$ in their average attention, which activates the second-layer attention head. Then the output is likely to predict the trigger itself, which will be an incorrect prediction most of the time.

We may thus consider $\hat{p}(k|X) = 1/N$ at this stage as well. We also consider a simplified architecture where the first layer attention only uses positional embeddings in the key-query matrix, and only token embeddings in the value-output matrix. In particular, we have $x_t = w_E(z_t)$. Lemma 5 then gives the following form for $W_K^1$ after one gradient step of step-size $\eta$:

$$W_K^1 = \frac{\eta}{N} \sum_{k=1}^N \mathbb{E}_X\left[\frac{1}{T}\sum_{t=1}^T w_U(k)^\top \Phi_2 x_t \cdot \frac{1}{t}\sum_{s=1}^t (\Phi_1 x_s)^\top W_K^2 x_T (p_s - \bar{p}_{1:t})p_t^\top|y = k\right]$$

$$- \frac{\eta}{N} \sum_{k=1}^N \mathbb{E}_X\left[\frac{1}{T}\sum_{t=1}^T w_U(k)^\top \Phi_2 x_t \cdot \frac{1}{t}\sum_{s=1}^t (\Phi_1 x_s)^\top W_K^2 x_T (p_s - \bar{p}_{1:t})p_t^\top\right]$$

$$- \frac{\eta}{N} \sum_{k=1}^N \mathbb{E}_X\left[w_U(k)^\top \Phi_2 \bar{x}_{1:T} \cdot \frac{1}{T}\sum_{t=1}^T \frac{1}{t}\sum_{s=1}^t (\Phi_1 x_s)^\top W_K^2 x_T (p_s - \bar{p}_{1:t})p_t^\top|y = k\right]$$

$$+ \frac{\eta}{N} \sum_{k=1}^N \mathbb{E}_X\left[w_U(k)^\top \Phi_2 \bar{x}_{1:T} \cdot \frac{1}{T}\sum_{t=1}^T \frac{1}{t}\sum_{s=1}^t (\Phi_1 x_s)^\top W_K^2 x_T (p_s - \bar{p}_{1:t})p_t^\top\right].$$

Note that since $W_O^2$ and $W_K^2$ already captured the desired associations at this stage, we have

$$w_U(k)^\top \Phi_2 x_t \approx \alpha\, \mathbb{1}\{z_t = k\} \quad \text{and} \quad (\Phi_1 x_s)^\top W_K^2 x_T \approx \alpha'\, \mathbb{1}\{z_s = z_T\},$$

for some $\alpha, \alpha' > 0$. Recall that in our model, we have $z_T = q$ with probability one ($q$ is the trigger token), and that $q$ only appears twice: once at position $t_q := t_o - 1 < T$ and once at position $T$. We then have, for any $t > 1$,

$$W_K^1 p_t \approx \frac{\eta \alpha \alpha'}{NTt} \sum_{k=1}^{N} (A_{t,k} - B_{t,k} - C_{t,k} + D_{t,k}),$$

with

$$A_{t,k} = \mathbb{E}[\mathbb{1}\{z_t=k\}\,\mathbb{1}\{t_q \leq t\}(p_{t_q} - \bar{p}_{1:t})|y=k] \tag{26}$$

$$B_{t,k} = \mathbb{E}[\mathbb{1}\{z_t=k\}\,\mathbb{1}\{t_q \leq t\}(p_{t_q} - \bar{p}_{1:t})] \tag{27}$$

$$C_{t,k} = \mathbb{E}[r_k\,\mathbb{1}\{t_q \leq t\}(p_{t_q} - \bar{p}_{1:t})|y=k] \tag{28}$$

$$D_{t,k} = \mathbb{E}[r_k\,\mathbb{1}\{t_q \leq t\}(p_{t_q} - \bar{p}_{1:t})], \tag{29}$$

where $r_k := \frac{1}{T}\sum_{t=1}^{T} \mathbb{1}\{z_t = k\}$. We have

$$A_{t,k} = \mathbb{E}[\mathbb{1}\{z_t=k\}(\mathbb{1}\{t_q = t-1\} + \mathbb{1}\{t_q \in [t-2] \cup \{t\}\})(p_{t_q} - \bar{p}_{1:t})|y=k]$$

$$= \mathbb{P}(t_q = t-1|y=k)(p_{t-1} - \bar{p}_{1:t}) + \frac{1}{N}\sum_{s\in[t-2]\cup\{t\}} \mathbb{P}(t_q = s|y=k)(p_s - \bar{p}_{1:t})$$

$$= \mathbb{P}(t_q = t-1)(p_{t-1} - \bar{p}_{1:t}) + \frac{1}{N}\sum_{s\in[t-2]\cup\{t\}} \mathbb{P}(t_q = s)(p_s - \bar{p}_{1:t})$$

$$= \mathbb{P}(t_q = t-1)(p_{t-1} - \bar{p}_{1:t}) + O\left(\frac{1}{N}\right),$$

since the distribution of $t_q$ is the same regardless of $y$. We proceed similarly for the other quantities and obtain the following:

$$B_{t,k} = O\left(\frac{1}{N}\right)$$

$$C_{t,k} = \frac{\mathbb{P}(t_q = t-1)}{T}(p_{t-1} - \bar{p}_{1:t}) + O\left(\frac{1}{N}\right)$$

$$D_{t,k} = O\left(\frac{1}{N}\right).$$

This yields the following associative memory behavior, for $t > 1$:

$$p_s^\top W_K^1 p_t \approx \frac{\eta \alpha \alpha'(T-1)}{T^2 t}\left\{\mathbb{P}(t_q = t-1)\left(\mathbb{1}\{s = t-1\} - \frac{1}{t}\,\mathbb{1}\{s \in [t]\}\right) + O\left(\frac{1}{N}\right)\right\},$$

which matches the desired "previous token head" behavior from (7) when $N$ is large. As in the case of $W_K^2$, we may then "saturate" the softmax by choosing a large enough step-size.

## C   Other Proofs

### C.1   Proof of Lemma 1

*Proof.* Recall the form of the cross-entropy loss for classification with $K$ classes:

$$\ell(y, \xi) = -\sum_{k=1}^{N} \mathbb{1}\{y = k\} \log \frac{e^{\xi_k}}{\sum_j e^{\xi_j}}.$$

Its derivatives take the form

$$\frac{\partial \ell}{\partial \xi_k}(y, \xi) = s(\xi)_k - \mathbb{1}\{y = k\},$$

with $s(\xi)_k = \frac{e^{\xi_k}}{\sum_j e^{\xi_j}}$ the softmax.

The gradient of $L$ is then given by

$$\nabla_W L(W) = \mathbb{E}_{(z,y)} \left[ \sum_{k=1}^{N} \frac{\partial \ell}{\partial \xi_k}(y, W_U W w_E(z)) \nabla_W (w_U(k)^\top W w_E(z)) \right]$$

$$= \mathbb{E}_{(z,y)} \left[ \sum_{k=1}^{N} (\hat{p}_W(k|z) - \mathbb{1}\{y = k\}) w_U(k) w_E(z)^\top \right]$$

$$= \sum_{k=1}^{N} \mathbb{E}_z [\mathbb{E}_y [(\hat{p}_W(k|z) - \mathbb{1}\{y = k\}) w_U(k) w_E(z)^\top \mid z]]$$

$$= \sum_{k=1}^{N} \mathbb{E}_z [(\hat{p}_W(k|z) - \mathbb{E}_y[\mathbb{1}\{y = k\}|z]) w_U(k) w_E(z)^\top],$$

which yields the desired result. □

## C.2 Proof of Lemma 2

*Proof.* Using similar steps as the proof of Lemma 1, we have

$$\nabla_W L(W) = \mathbb{E}_{(x,y)} \left[ \sum_{k=1}^{N} \frac{\partial \ell}{\partial \xi_k}(y, W_U W x) \nabla_W (w_U(k)^\top W x) \right]$$

$$= \mathbb{E}_{(x,y)} \left[ \sum_{k=1}^{N} (\hat{p}_W(k|x) - \mathbb{1}\{y = k\}) w_U(k) x^\top \right]$$

$$= \sum_{k=1}^{N} w_U(k) \, \mathbb{E}_x [\hat{p}_W(k|x) x]^\top - \sum_{k=1}^{N} \mathbb{E}_y [\mathbb{1}\{y = k\} w_U(k) \, \mathbb{E}[x|y]^\top]$$

$$= \sum_{k=1}^{N} w_U(k) \, \mathbb{E}_x [\hat{p}_W(k|x) x]^\top - \sum_{k,j=1}^{N} p(y = j) \, \mathbb{1}\{j = k\} w_U(k) \, \mathbb{E}[x|y = j]^\top$$

$$= \sum_{k=1}^{N} p(y = k) w_U(k) (\hat{\mu}_k - \mu_k)^\top,$$

with $\hat{\mu}_k = p(y = k)^{-1} \, \mathbb{E}_x [\hat{p}_W(k|x) x]$ and $\mu_k = \mathbb{E}[x|y = k]$. □

## C.3 Proof of Lemma 4

*Proof.* To better isolate the role of keys from values, we denote the keys that are fed into the matrix $W$ by $Z = [z_1, \ldots, z_T] \in \mathbb{R}^{d \times T}$, while the query is simply $x_T$. In practice we have $Z = X$, and both are superpositions of potentially multiple embeddings (if $W$ is part of the second attention layer, these are the token embedding, positional embedding, and the output of the first attention layer).

The gradient of the loss at $W = 0$ writes:

$$\nabla_W L(W)\big|_{W=0} = \mathbb{E}_{(X,Z,y)} \left[ \sum_{k=1}^{N} \frac{\partial \ell}{\partial \xi_k}(y, \xi) \cdot \nabla_W (w_U(k)^\top \Phi_2 X \sigma(Z^\top W x_T))\big|_{W=0} \right] \quad (30)$$

$$= \mathbb{E}_{(X,Z,y)} \left[ \sum_{k=1}^{N} (\hat{p}_W(k|X, Z) - \mathbb{1}\{y = k\}) \cdot \nabla_W (w_U(k)^\top \Phi_2 X \sigma(Z^\top W x_T))\big|_{W=0} \right]. \quad (31)$$

We have

$$\nabla_W (w_U(k)^\top \Phi_2 X \sigma(Z^\top W x_T))\big|_{W=0} = \sum_{t=1}^{T} w_U(k)^\top \Phi_2 x_t \cdot \nabla_W (\sigma(Z^\top W x_T)_t)$$

$$= \frac{1}{T} \sum_{t=1}^{T} w_U(k)^\top \Phi_2 x_t \cdot (z_t - \bar{z}_{1:T}) x_T^\top,$$

where $\bar{z}_{1:T} = \frac{1}{T}\sum_t z_t$, and we used the fact that

$$\frac{\partial}{\partial u_s}\sigma(u)_t\big|_{u=0} = \frac{1}{T}\mathbb{1}\{t=s\} - \frac{1}{T^2}. \tag{32}$$

The gradient (31) now writes

$$\nabla_W L(W)\big|_{W=0} = \sum_{k=1}^N \mathbb{E}_{(X,Z)}[(\hat{p}_W(k|X,Z) - \mathbb{1}\{y=k\})\frac{1}{T}\sum_{t=1}^T w_U(k)^\top \Phi_2 x_t \cdot (z_t - \bar{z}_{1:T})x_T^\top],$$

and the result follows. $\qquad\square$

### C.4 Proof of Lemma 5

*Proof.* The linearization of the second layer softmax around zero takes the following form:

$$\bar{\sigma}(Z^\top W_2 x_T)_t = \frac{1}{T}(1 + z_t^\top W_2 x_T - \bar{z}_{1:T}^\top W_2 x_T),$$

with $z_t = \sum_{s=1}^t \Phi_1 x_s \sigma(p_{1:t}^\top W p_t)_s$ the output of the first attention layer.

$$\begin{aligned}
\xi_k &= \sum_{t=1}^T w_U(k)^\top \Phi_2 x_t \bar{\sigma}(Z^\top W_2 x_T) \\
&= \frac{1}{T}\sum_{t=1}^T w_U(k)^\top \Phi_2 x_t + \frac{1}{T}\sum_{t=1}^T w_U(k)^\top \Phi_2 x_t \sum_{s=1}^t (\Phi_1 x_s)^\top W_2 x_T \sigma(p_{1:t}^\top W p_t)_s \\
&\quad - w_u(k)^\top \Phi_2 \bar{x}_{1:T} \cdot \frac{1}{T}\sum_{t=1}^T \sum_{s=1}^t (\Phi_1 x_s)^\top W_2 x_T \sigma(p_{1:t}^\top W p_t)_s.
\end{aligned}$$

Then,

$$\nabla_W L(W)\big|_{W=0} \tag{33}$$

$$= \mathbb{E}_{(X,y)}\left[\sum_{k=1}^N \frac{\partial \ell}{\partial \xi_k}(y,\xi) \cdot \nabla_W \xi_k\big|_{W=0}\right] \tag{34}$$

$$= \mathbb{E}_{(X,y)}\left[\sum_{k=1}^N \frac{\partial \ell}{\partial \xi_k}(y,\xi)\frac{1}{T}\sum_{t=1}^T w_U(k)^\top \Phi_2 x_t \frac{1}{t}\sum_{s=1}^t (\Phi_1 x_s)^\top W_2 x_T (p_s - \bar{p}_{1:t})p_t^\top\right] \tag{35}$$

$$\quad - \mathbb{E}_{(X,y)}\left[\sum_{k=1}^N \frac{\partial \ell}{\partial \xi_k}(y,\xi)w_U(k)^\top \Phi_2 \bar{x}_{1:T} \cdot \frac{1}{T}\sum_{t=1}^T \frac{1}{t}\sum_{s=1}^t (\Phi_1 x_s)^\top W_2 x_T (p_s - \bar{p}_{1:t})p_t^\top\right], \tag{36}$$

using (32). The result follows by using $\frac{\partial \ell}{\partial \xi_k}(y,\xi) = \hat{p}(k|\xi) - \mathbb{1}\{y=k\}$.

$\qquad\square$

## D   Beyond our Simplified Architecture

While the simplified architecture presented in Section 4.2 is sufficient to capture the desired induction behavior for our bigram task, transformer architectures used in practice typically involve more components, as well as more heads and layers. In this section, we discuss how our memory viewpoint extends to such architectures.

**Factorizations.** In practice, transformers typically involve products of matrices, potentially with a low-rank bottleneck. For instance, our key-query matrices $W_K$ should instead be considered as a product $W_K^\top W_Q$, and the output-value matrices $W_O$ and $W_V$ are typically jointly optimized.

Consider an associative memory of the form:

$$W_* = \sum_i y_i x_i^\top \in \mathbb{R}^{d \times d},$$

where $(x_i)_i$ and $(y_i)_i$ are appropriate collections of near-orthonormal embeddings.

We now argue that a similar associative memory can be achieved with the factorization $W = \frac{d}{2d'} UV$, where $U \in \mathbb{R}^{d \times d'}$, $V \in \mathbb{R}^{d' \times d}$ with $d' \leq d$ (for instance $d'$ could be the dimension of attention heads), are given by:[3]

$$U = U_0 + \sum_i y_i (V_0 x_i)^\top$$

$$V = V_0 + \sum_i (U_0^\top y_i) x_i^\top,$$

where $U_0$ and $V_0$ are random matrices with $\mathcal{N}(0, \frac{1}{d})$ entries. These matrices are similar to those that would arise from a single gradient step individually on $U$ and $V$ from initializations $U_0$ and $V_0$, as in Lemma 1. To see why $W$ behaves like $W_*$, note that we have

$$UV = U_0 V_0 + \sum_i y_i (V_0 x_i)^\top V_0 + \sum_i (U_0^\top y_i) x_i^\top + \sum_{i,j} y_i (V_0 x_i)^\top (U_0^\top y_j) x_j^\top.$$

It is also easy to check using central limit arguments (similar to remapping in Appendix A) that $\tilde{x}_i := \sqrt{\frac{d}{d'}} V_0 x_i \in \mathbb{R}^{d'}$ and $\tilde{y}_i := \sqrt{\frac{d}{d'}} U_0^\top y_i$ are all nearly-orthonormal embeddings. Thus, we have

$$2 y_k^\top W x_l = \tilde{y}_k^\top \tilde{x}_l + \sum_i y_k^\top y_i \tilde{x}_i^\top \tilde{x}_l + \sum_i \tilde{y}_k^\top \tilde{y}_i x_i^\top x_l + \sum_{i,j} y_k^\top y_i \tilde{x}_i^\top \tilde{y}_j x_j^\top x_l$$

$$\approx 0 + \mathbb{1}\{k = l\} + \mathbb{1}\{k = l\} + 0,$$

where the first and last term vanish due to the cross-terms $\tilde{y}_i^\top \tilde{x}_{i'}$ which vanish for any $i, i'$. Thus, $W$ and $W_*$ encode the same associations, when $d$ and $d'$ are large enough to ensure near-orthogonality.

**Layer-normalization.** Normalization layers [4, 60] are typically used in transformers to improve training stability [52], and are applied on each token representation, either after [52] or before [6] each block. It may be seen as an operation of the form[4]

$$LN(x) = \frac{x}{\|x\|},$$

applied to the input or output of a given block.

In order to obtain a basic understanding of the role of layer-norm in our associative memory setup, we may consider the setup Lemma 1, with a normalization applied after the linear operation, leading to the population loss:

$$L(W) = \mathbb{E}_{(x,y) \sim p}[\ell(y, W_U LN(Wx))]. \tag{37}$$

The gradients then take the form

$$\nabla_W L(W) = \sum_{k=1}^N \mathbb{E}_x \left[ \frac{\hat{p}_W(y = k|x) - p(y = k|x)}{\|Wx\|} \left( I - \frac{(Wx)(Wx)^\top}{\|Wx\|^2} \right) w_U(k) x^\top \right]. \tag{38}$$

---

[3]When $d'$ is the head dimension, the $\frac{d}{d'}$ scaling can be interpreted as the correct multiplier to use in attention logits, which plays a similar role to the $\frac{1}{d'}$ multiplier in the $\mu$P scaling [59], for our setup where the variance of the random entries of input embeddings is $1/d$ instead of 1 as in [59].

[4]RMSNorm [60] would use the variance instead of the norm in the denominator, leading to an additional $\sqrt{d}$ factor in the numerator. Here we use the norm, which is more natural when embeddings have near-unit norm, in contrast to the $\approx \sqrt{d}$ norm for the standard parameterization.

This illustrates that in addition to weighting the updates on class $k$ by the prediction error $\hat{p}(k|x) - p(k|x)$, the updates are also projected on the orthogonal of the $Wx$ direction. This means that an update on the direction $w_U(k)x^\top$ will occur only to the extent that $Wx$ is not already aligned with $w_U(k)$. Thus, if an association is "stored" once, so that $Wx \approx w_U(y)$, layer-norm will avoid further updating $W$ in that direction, hopefully avoiding norms that grow too much, and also encouraging frequent and infrequent tokens to be weighted similarly in the final memory (see also [7] for more discussion on this).

Note that at random initialization $Wx$ is nearly orthogonal to any $w_U(k)$, so that layer-norm only starts playing a significant role later in training, and does not affect our theoretical analysis based on single gradient steps.

**MLP blocks.** If we denote by $(u_i)_i$ and $(v_i)_i$ collections of near-orthonormal input and output embeddings, an MLP block may encode associations $(i, j) \in \mathcal{M}$ as follows:

$$F(x) = \sum_{(i,j)\in\mathcal{M}} v_j \sigma(u_i^\top x - b),$$

where $\sigma$ is a non-linear activation and $b$ a bias term. Then, if one assumes that $u_i^\top u_j \leq b$ for $i \neq j$ and $\sigma(t) = 0$ for $t < 0$, then this can help filter out noise that arises from near-orthogonality, and may then lead to additional storage capacity, at the cost of additional computation (see, *e.g.*, [7, 30]).

An additional benefit of MLP layers discussed in Section 4 is that they may encode many-to-many associations, which is useful when multiple embeddings are in the residual stream and need to be considered jointly (*e.g.*, a subject and a relation in a factual recall task [35]). This may be achieved, for instance, by considering embeddings $u_\mathcal{I} = \frac{1}{\sqrt{|\mathcal{I}|}} \sum_{i\in\mathcal{I}} u_i$, where $\mathcal{I}$ are sets of bounded size (*e.g.*, as obtained using layer-norm over a residual stream). Then, assuming the $u_i$ are nearly-orthonormal, we have $u_\mathcal{I}^\top u_\mathcal{I} \approx 1$ while $u_\mathcal{I}^\top u_{\mathcal{I}'} \lesssim 1 - \delta$ if $\mathcal{I} = \mathcal{I}'$, for some $\delta$ that depends on the maximal cardinality of these sets. Although $1 - \delta$ is no-longer vanishingly small, defining

$$F(x) = \sum_{(\mathcal{I},\mathcal{J})\in\mathcal{M}} v_\mathcal{J} \sigma(u_\mathcal{I}^\top x - b),$$

the non-linearity may still succeed at filtering out any $\mathcal{I}$ that does not correspond to the query set in $x$. We leave the question of how such non-linear associative memories may arise from training dynamics to future work.

**Multiple heads and layers.** We remark that our view of weights as associative memories applies to any parameter other than embedding/unembedding layers, and thus naturally extends to multiple heads (using the low-rank factorizations described above) and multi-layer models.

It is important to note, however, that the redundancy introduced by having more heads and layers makes it more challenging to identify which layer/head/weight will learn certain associations (see, *e.g.*, Figure 11 in Appendix E). This is in contrast to our simplified architecture of Section 4.2, where we may identify the role of each matrix (up to some possible redundancy when using the feed-forward layer $W_F$). In practice, mechanisms may appear in different heads/layers across different training runs, which makes interpretability more challenging, and typically requires some causal identification techniques, such as mediation analysis [35, 54].

# E   Experiment Details and Additional Experiments

In this section, we present additional details on the experiments, as well as additional results.

**Computing setup.** We use Pytorch and each run uses a single GPU, along with 60 CPU cores for real-time data generation. We will make our code available upon publication.

**Hyperparameters.** We now provide the hyperparameters used in each figure. The SGD step-size is denoted $\eta$. We fix the momentum parameter to 0.9 and the weight decay parameter to $10^{-4}$. $U$ denotes the uniform distribution over $[N]$.

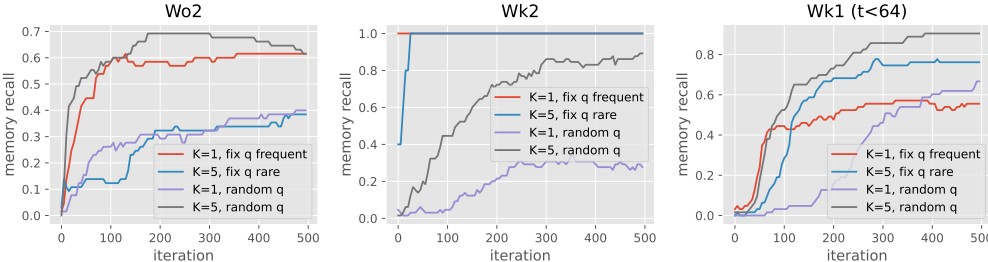

Figure 5: Memory recall probes for the setting of Figure 4(left).

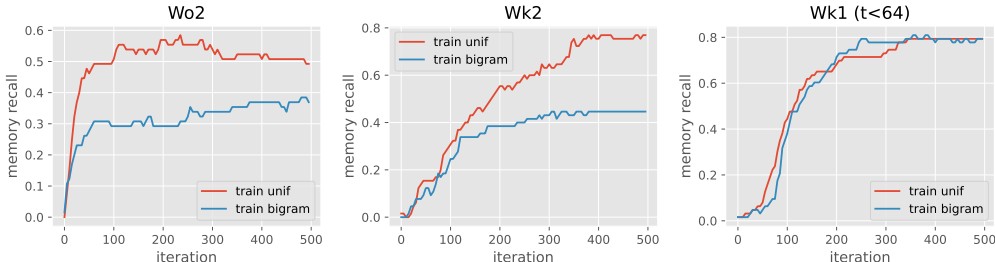

Figure 6: Memory recall probes for the setting of Figure 4(center).

- Figure 2: $K = 3$, $\pi_q = \pi_u$ (random triggers) or $Q$ is the $K$ most likely elements of $\pi_u$, $\pi_o = U$, $d = 128$, $d_{hidden} = 4 \times 128$ (hidden dimension of the feed-forward MLPs), $\eta = 0.2$.

- Figure 3: $K = 5$, $\pi_q = \pi_u$ (random triggers), $\pi_o = U$, $d = 128$, $\eta = 0.2$.

- Figure 4(left) and Figure 5: $\pi_o = U$, $d = 128$, $\eta = 1$. For random triggers we use $\pi_q = \pi_u$. For $K = 1$ with fixed frequent trigger, the only trigger is the most probable token according to $\pi_u$, while for $K = 5$ with fixed rare triggers, the five triggers are the 6-th to 10-th most probable tokens according to $\pi_u$.

- Figure 4(center): $K = 3$, $\pi_q = \pi_u$ (random triggers), $\pi_o = U$ or $\pi_o = \pi_b$ (conditioned on the trigger), $d = 128$, $\eta = 1$.

- Figure 4(right): $K = 3$, $\pi_q = \pi_u$ (random triggers), $\pi_o = U$, $d = 128$, $\eta = 1$.

**Memory recall probes and data-distributional properties.** Figure 5 and Figure 6 show the evolution of the different memory probes for the settings considered in Figure 4(left,center). Figure 5 highlights that associative memories for the induction head are slower to learn when using few triggers (small $K$), rare fixed triggers, or random triggers (note that the probe for $W_K^2$ with fixed triggers only shows recall accuracy on the set of triggers $Q$, which is an easier task). Figure 6

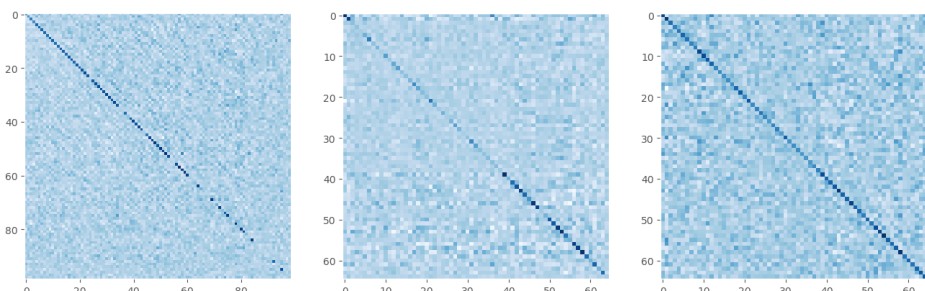

Figure 7: Visualization of the weights $W_K^1$ (left), $W_K^2$ (center), and $W_O^2$ (right) after training with random triggers, $K = 3$, $\pi_q = \pi_u$, $\pi_o = U$. For each of these weight matrices $W$, if we write the corresponding target memory in (7) as $W_* = \sum_i v_i u_i^\top$ with appropriate embeddings $(u_i)_i$ and $(v_i)_i$ (for instance $u_t = p_{t-1}$ and $v_t = p_t$ for $W_K^1$ on the left), and we show all values $v_j^\top W u_i$.

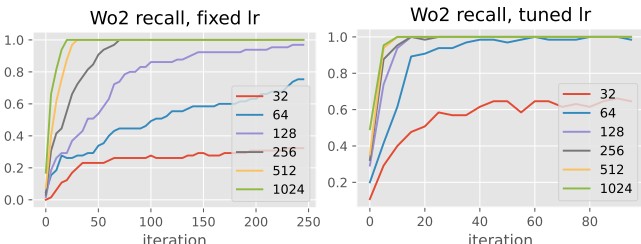

Figure 8: Effect of dimension on learning $W_O^2$ alone, with fixed or tuned learning rate.

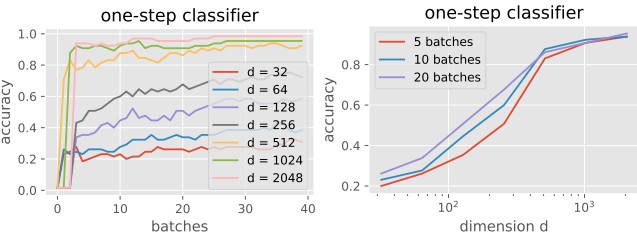

Figure 9: Accuracy of one-step estimate of $W_O^2$ with varying dimension and number of batches used for computing expectations. Each batch consists of 32 sequences of 256 tokens for a total of 8 192 tokens, with $K = 5$ random triggers and uniform outputs.

shows that using uniform output tokens can lead to better fitting of $W_O^2$ and $W_K^2$ compared to using output tokens sampled from bigrams. In addition to the increased diversity when using uniform outputs, this may also be due to the fact that bigram outputs are already well predicted using global statistics with just the feed-forward layer, hence the gradient signal on such well-predicted tokens may not propagate through the induction head mechanism. In contrast, the recall accuracy for $W_K^1$ is comparable for both settings, since the previous token head is useful at all positions regardless of the output token distribution.

**Visualizing memories.** Figure 7 shows visualizations of the associative memory behaviors after training. We see that diagonal elements dominate in the plots, which corresponds to correct associations lead to high 'memory recall'. Nonetheless, we see that some of the diagonal elements are weaker than others, particularly for late positions in $W_K^1$, and for some of the trigger tokens in $W_K^2$, while the diagonal for $W_O^2$ seems to be roughly uniform. We note that characters corresponding to capital letters have token index 13 to 38, while lowercase letters have index 39 to 64. The association patterns found in $W_K^2$ then seem related to frequencies of appearance of triggers, whereby capital letters appear less frequently in the data, and are also less frequently chosen as triggers, compared to lowercase letters. Similarly, since the first occurrence of triggers is typically early in a sequence, it is natural that $W_K^1$ learns stronger associations at earlier positions. In contrast, diagonal elements for $W_O^2$ are nearly uniform, which agrees with the fact that output tokens are sampled uniformly in this setup. We refer to the follow-up work [7] for an analysis of how data frequencies affect association strength in such associative memories.

**Effect of dimension.** Recall that our study of associative memories with random embeddings requires large dimension $d$ in order to ensure near-orthogonality, and thus store input-output pairs more effectively. In Figure 8, we evaluate the recall accuracy for $W_O^2$ for varying dimension, when training it by itself, and only on the output tokens (as in Figure 3). We see that higher dimension leads to faster learning of the memory, in particular $d = 128$ seems sufficient for fast learning after just a few iterations with a tuned learning rate. If the learning rate isn't tuned, we notice that there is a further slowdown for low dimension, is likely due to issues with the fact that our experiments use the standard parameterization of neural networks at initialization, rather than maximal update parameterizations [58]. Note that learning $W_O^2$ alone is a convex optimization problem, and we hypothesize that higher dimension makes the problem better conditioned, and hence easier to learn. In Figure 9, we show "one-step" recall accuracies for classifying output tokens from the average

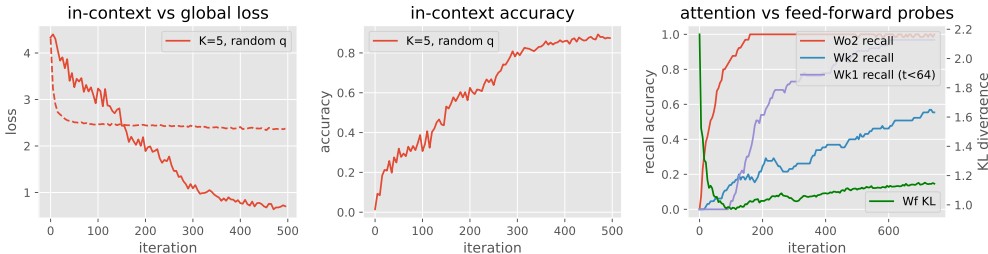

Figure 10: Training of a more realistic architecture with (i) ReLU MLP instead of linear layer for the second feed-forward layer, (ii) all parameters trained, including embeddings, (iii) pre-layer normalization. The loss, in-context accuracy and memory recall probes are similar to the simplified architecture (see, *e.g.*, Figure 4).

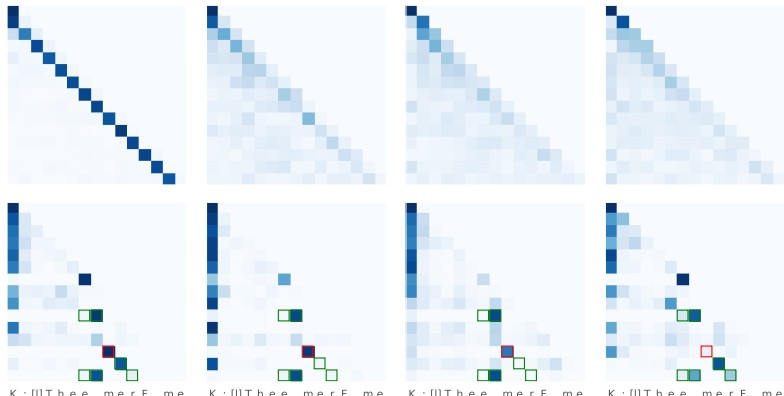

Figure 11: Attention maps for a two-layer model with 4 attention heads. In the first layer (top), the previous token mechanism is mostly achieved by one of the four heads, while the induction behavior at the second layer (bottom) is distributed across the different heads.

attention input to $W_O^2$, given by

$$R_1 = \frac{1}{N} \sum_{k=1}^{N} \mathbb{1} \left\{ k = \arg \max_{k'} (W_V^2 w_E(k'))^\top (\mu_k - \mu) \right\},$$

where $\mu_k = \mathbb{E}[x | y = k]$ and $\mu = \mathbb{E}[x]$, for $x = \frac{1}{t} \sum_{s=1}^{t} W_V^2 w_E(z_s)$ and $y = z_{t+1}$, when $z_t$ is a trigger token after its first occurrence. Expectations are computed over batches of data of varying sizes and in different dimensions. We call this "one-step" since it is related to the classifier obtained after performing a single gradient step on $W_O^2$ from zero initialization (see Lemma 2 and Appendix B.3.1). The plots illustrate that this simple one-step model is already able to extract relevant signal from the noisy average attention, after a handful of batches of data, corresponding to tens of thousands of tokens, and that this gets easier as the dimension increases.

**More complex architectures.** Figure 10 shows training behavior for a more complex model than the simplified one considered in Section 5, namely where we train all parameters, replace the linear $W_F$ feedforward layer by a two-layer MLP, and were (pre-)layer-normalization is added. Despite these changes, we see similar behavior for the memory recall probes (which now involve embeddings that may change over time), suggesting that the model is still identifying the same memory associations, despite the additional redundancies in parameters and modified training dynamics.

Figure 11 shows the attention maps obtained when training a multi-head version of our two-layer model, with four attention heads per layer. We see that the redundancy of multiple heads creates difficulties in identifiability: only one of the first layer heads learns the previous token behavior, while the induction behavior is shared across different heads at the second layer. This illustrates the challenges of interpretability in the presence of redundant models, which then require additional work to identify which of the layers and heads are performing a given behavior, *e.g.*, through interventions and causal mediation analysis [35, 54].

