# OpenReview forum: "Birth of a Transformer: A Memory Viewpoint"
_NeurIPS.cc/2023/Conference — NeurIPS 2023 spotlight_

### Official Review · Reviewer_k5yo · 2023-07-01

**Soundness:** 2 fair
**Presentation:** 3 good
**Contribution:** 2 fair
**Rating:** 5
**Confidence:** 4

**Summary:**

The paper studies in detail two-layer transformers and extend the setting of pure associative recall based on data in-context with mixing these tasks with tasks coming from a global birgram model. They then study these two-layer Transformers by freezing layers and probing.

**Strengths:**

The paper studies an interesting problem i.e. in-context learning within Transformers, that is believed to be an important characteristic of language models in general. Based on recent work, they aim to mechanistically interpret / reverse engineer the two layer Transformer which in my opinion is an interesting direction to better understand and get intuition on how Transformers work.

I like the mixed bigram data setup that the authors propose and study and I share the opinion of the authors that the tension between storing knowledge and quick adaptation / learning from in-context is very interesting and not explored. Progress in this direction is important.

**Weaknesses:**

It feels that the paper is very rushed. I think therefore that the presentation can be vastly improved as well as results and analyses refined.
Furthermore, I think that the proposed method of freezing the vast majority of layers in the network leads to very biased results and therefore results which might not hold when training on all weights.

A couple of design decisions seem also quite arbitrary and it would be I think the job of the paper to justify and investigate them. See Questions. Therefore, I think that the paper is very promising but can and should be improved.

Minor things / Language:
Key, Query, Value Matrix must not be square and the positional encodings are usually not randomly initialized. Please at least comment on these design decisions. Also you do not use layer norm, I would also expect at least a comment on this. Please use LaTex in your Figures, quite hard to read the legends etc (maybe different background color as well?). The sum in Figure 2 over the positional encodings should be over t? Please describe more precisely Figure 3&4 in the caption. These are not concise i.e. it was not possible for me to understand what you are plotting what the Figures show.  Also please increase the font size.

At least one interesting citation is missing: https://arxiv.org/abs/2212.07677
The paper studies in details 1) single and two layer Transformer and provide evidence about copying in the first layer and 2) in-context learning in the second layer by gradient descent. This is, I believe, equivalent to the Hebbian-rule / associative recall with orthogonal inputs.

**Questions:**

In general I think the papers focus on the tension between memory and in-context learning is great.
Nevertheless, I think that a couple of very interesting experiments, ablations and analyses about design decisions are missing.

1) Can you provide analyses in both extremes i.e. when there are only "triggers" i.e. when all sequences can be learned by quick associative learning and the other way around. This should lead to quite different circuits that can be contrasted against each other.

2) Why do you include a single (linear) feedforward layer in the architecture? The problem should be solvable without (as studied in the induction head paper). If you want to include this layer, then I think the right thing to do would be to include it also after the second layer.

3) Why are you so heavily relying on the freezing of some layers? I actually ran your experiments (with only triggers) and I believe that my matrices do actually show very different behavior. I think that the freezing of the layers, forces you to focus on the superposition too much. If you wouldn't freeze your vocabulary and the input matrix, the model could (and I see this in my experiments) actually produce embeddings which leave token memory free to use for layer computations i.e. essentially concatenate x_i = w_E(z_i) + w_P(i) \approx [z_i, p_i] since the positional encoding is basically alternating 0 and 1 after some dimension.  By that, the circuits and mechanism that you find could be very different and therefore not stable wrt. to design decisions.

4) Building on 3): I am quite convinced that all design decisions will change the circuits and mechanism of how the two layers solve your problem. Therefore I think it is very interesting and actually needed to investigate these and make suggestion what these design decisions control. E.g. Add or concat frozen/learnable random/sinusoidal PEs, control the dimension of the vocab and the token dimension, include feedforward layer, etc.

5) The theoretical insights on Learning Dynamics seem a bit vague and less supported by empirical study (didnt look in the appendix). As I understand, in-context learning kicks in later in training, so why study on the random init? At least comment on this shortcoming, if I understood this correctly.

Minor things:

5) Can you please better your probing experiments, how exactly do you do that? Which vectors are probed?
6) What do you mean with discrete tokens line 81?
7)  Equation 7 - its confusing for me why exactly these weights recover the induction head mechanism. Please elobatore, I can somehow see it but please make this more rigorous.
8) Please explain a bit better your data generation and the statistics. How large is the possibility of conflicts i.e. a -> based on global knowledge and -> based on in-context.
9) Figure 4 left. Why cant the in-context learning loss go to zero? Is it because of the conflicts? Please make the Figures a bit less crowded, hard to see all the lines. Would be great to have scaling plots wrt to K imo.









**Limitations:**

I dont think the limitations of design decisions are well discussed or investigated. See Questions.

---

> ### Author Rebuttal · Authors · 2023-08-09
>
> Thank you for the detailed review and for the interest to try out our code! We hope our response can provide more perspective about the motivations behind our work, and may help you reconsider your score.
>
> *freezing layers*
>
> See our general response. In particular, we chose to simplify the architecture as much as possible to have a simpler setup for a clear study. We also found empirically that training all parameters, even when adding layer-norm and MLPs, still leads to correct memory recall associations.
>
>
> *Minor Things*
>
> * "Key, Query, Value...": with a single head, V needs to be squared, but we agree that Q/K need not be squared. We will include the case of separate and potentially non-square Q/K matrices, which still can lead to the desired memories for the product $W_Q^\top W_K$.
> * positional encodings: to our knowledge, when using learned positional encodings, these are randomly initialized.
> * layer-norm: see general response (it still works!)
> * Figures: thanks for your suggestions! We will improve this in the next version
> * sum in Figure 2: we use the variable $s$ instead of $t$ (as used in Eq. 7) in the sum to avoid confusion with the $t$ of the tokens below.
>
>
> *Missing reference*
>
> Thanks for pointing out this very relevant paper! We will cite it and compare to it in the next version.
>
> The mechanism they describe is indeed similar to ours, and to induction heads in general. However, their construction relies on idealized choices of weights such as the identity matrix (see A.1 in their paper), which could easily be quite different from the weights learned by training, particularly in a setup with discrete data like ours. In contrast, our work (i) provides a precise description of what the weights look like with gradient updates, (ii) verifies this form empirically through memory probes, and (iii) shows theoretically that gradient dynamics on our bigram task indeed recovers these associative memories.
>
>
> *"..analyses in both extremes..."*
>
> Our model already covers all cases ($K=0$ means no triggers, $K=N$ means all tokens are triggers). For $K=0$, the model just doesn't learn the induction mechanism since the gradients to the weights of the attention blocks are essentially dominated by noise. For $K \geq 1$, the same induction head mechanism is learned as long as triggers appear in the data (which is why our experiments consider triggers that are frequent tokens). The only difference is that $W_K^2$ in eq.7 only learns associations for trigger tokens that appear in the data. This is described at the end of Section 4.2, but we will clarify it further.
>
> *"..single (linear) feedforward layer..."*
>
> You are right that global bigrams may be learned without feedforward layers (as in the induction head work), however that would require learning embeddings. With fixed random embeddings, it is much harder to encode global bigrams without feed-forward layers (see response to hHJQ).
> We prefer encoding this in a linear feedforward layer because its training fits the associative memory viewpoint, as for all internal weight matrices, and might thus capture how other feed-forward layers may store global knowledge in larger networks. Input/output embedding layers likely exhibit quite different learning dynamics that go beyond the message of our work. As a side note, even when training all parameters including embeddings, we observed that the KL probe for $W_F$ still decreases quite quickly, which suggests that gradient dynamics may prefer storying global bigrams in the feed-forward layer as opposed to embedding layers.
>
> *"Why are you so heavily relying on the freezing... I actually ran your experiments ..."*
>
> Thank you for taking the time to run our code!
>
> We tried training all layers including embeddings, and found that the recall probes actually reach perfect recall accuracy (see general response), even when using only triggers (i.e. using the option `--data_args.k 65`). Thus, we believe the model still learns the same mechanism.
>
> If however you are using fixed sinusoidal positional encodings, then the first layer might end up learning something different than eq. 7 for $W_K^1$, since you no longer have the near-orthogonality behavior, and in fact you may get a previous-token head behavior without the need to store outer products for all t thanks to the sinusoidal structure (but the other layers still do get behave like eq. 7). This relies on specific embedding structures that depart from our memory viewpoint based on near-orthogonality, which is why we did not include sinusoidal PEs in our paper, but we agree it is an interesting direction!
>
>
> *"The theoretical insights..."
>
> The detailed analysis of how the induction head mechanism emerges is deferred to Appendix B.3 as it is quite technically involved. We instead focused on the key ingredient behind the proofs, which is simpler to state, and is about how gradient updates on a lot of data can filter out irrelevant parts elements of input superpositions.
>
> Regarding "later in training": we show that if you learn sequentially $W_O^2$, then $W_K^2$, then $W_K^1$, each with a single gradient step from its initialization and on enough data, then you *do* learn the in-context learning/induction head mechanism (i.e., there is no shortcoming). The key point is "enough data" and also what kind of data: learning the global bigrams is typically very fast because there is a lot of such bigrams and the signal is quite "clean" and accessible from the current token's residual stream. In contrast, for the induction head, say, to learn $W_O^2$, this requires finding signal among *all tokens* seen so far, which is much harder and requires more data (in addition to the fact that trigger-output pairs are often less frequent overall compared to global bigrams).
> We'd be happy to include these intuitions in the paper, if it is accepted.
>
> Thank you for your other comments and suggestions, we will happily incorporate them in the updated version.

---

> > ### Comment · Reviewer_k5yo · 2023-08-12
> > **Thank you**
> >
> > Thank you for your thorough and thoughtful response. I will increase my rather low score accordingly. Thanks again

---

> > > ### Author Response · Authors · 2023-08-15
> > > **Thank you**
> > >
> > > Thank you for your message and for increasing your score, we appreciate it!

---

### Official Review · Reviewer_PfRr · 2023-07-02

**Soundness:** 3 good
**Presentation:** 2 fair
**Contribution:** 3 good
**Rating:** 7
**Confidence:** 3

**Summary:**

This paper provides a detailed analysis of how in-context learning behavior emerges in a simplified version of the transformer architecture on a toy task. This work can provide important insight into how in-context learning emerges in LLMs. The toy task is this: given a sequence of tokens of the form $\ldots, a, b, \ldots, a$, predict $b$. The token $a$ is a "trigger" token. If token $b$ (the "output" token) comes after the first occurrence of $a$, then the model must predict $b$ after the second occurrence of $a$. The model is expected to use in-context learning to predict the correct $b$. The rest of the sequence follows a bigram language model distribution. The authors zoom in on two aspects of the model: the "induction head" mechanism that implements in-context learning, and the "associative memory" mechanism in the key/query weight matrices that allows the model to learn the global bigram distribution. They give mathematical justification for the emergence of both mechanisms in their simplified transformer model trained on infinite data. They provide experimental results that provide evidence that these mechanisms are learned in a particular order, and that learnability depends on attributes of the training data distribution, such as the number of trigger tokens per sequence, or whether the trigger token types are randomized.

**Strengths:**

This paper provides a very useful case study of how in-context learning behavior emerges in transformers. In-context learning is a hot topic, and it is important to understand architecturally how this behavior emerges, as well as the conditions that are conducive to it. This paper is a good step towards better understanding this phenomenon. I think their bigram task is an appropriate choice of case study. The authors analyze two mechanisms and provide mathematical and empirical justification for both.

Originality: Good. I think the choice of task is very useful for the purposes of studying in-context learning.

Quality: The mathematical analysis and experimental design appear to be sound.

Clarity: The paper seems to be well-contextualized with respect to previous work.

Significance: High. After reading this paper, I feel I have a better grasp on how in-context learning works in the transformer architecture, and this has important implications for any NLP applications that use transformers.

**Weaknesses:**

**Edit:** I have read the rebuttal, and it addressed my biggest concerns.

I have two major criticisms of this paper, which is why I did not immediately assign it a higher score:

1. The transformer architecture used in this paper is drastically simplified from real transformers used in LLMs. There is no layer normalization or dropout, and the transformer only has 2 layers. They use linear layers instead of feedforward layers. The key and value matrices are merged into one matrix. Many of the parameters are frozen during experiments and during gradient analysis. I think the paper requires a much more detailed, readable discussion justifying why the authors expect their analysis on this simplified transformer model to carry over to real transformers. I think there is some discussion scattered throughout the paper, but if so, I think it needs to be organized more clearly. I think the results are useful regardless of these simplifying assumptions, but the authors need to talk about this more.
2. Clarity. This was a very difficult paper to read, and this negatively impacted my ability to interpret the results. As I explain in more detail in the Questions section, a recurring issue I had while reading this paper was that information is frequently presented in *reverse* order; I often needed to read several lines, paragraphs, or sections ahead in order to clarify something I was confused about. I think the whole paper would benefit from a round of editing that alleviates these issues. See the Questions section for more details.

**Questions:**

Most important questions:

1. 77, 90, 93, 97: Does the decision to ignore layer normalization make the findings of this paper less applicable to real transformers and LLMs? Same question for using only a single head. What about dropout? Same question about using linear transformations instead of feed-forward networks.
1. Fig 1: I had a very hard time understanding this figure, and it's essential for understanding the paper. I don't know what Attn1 and Attn2 are supposed to signify. In Layer 1 column 2, $w_1(a)$ and $w_E(b)$ aren't the only values in superposition there, right? It would also have $p_t$. It's not possible to completely isolate $w_E(b)$ from $p_t$, because they are in superposition in Layer 0, right? I think it would really help to give a symbolic representation of the query, key, and value functions at each layer. If I understand correctly, the inputs at Layer 1 are essentially $(x_t, t)$, where $x_t$ is the input token at timestep $t$. Then $\mathrm{query}(x_t, t) = t-1$ (assuming you can compute this with a linear layer), $\mathrm{key}(x_t, t) = t$, and $\mathrm{value}(x_t, t) = x_t$. This adds $x_{t-1}$ to the inputs of Layer 2. Then in Layer 2, $\mathrm{query}(x_t, t, x_{t-1}) = x_t$, $\mathrm{key}(x_t, t, x_{t-1}) = x_{t-1}$, and $\mathrm{value}(x_t, t, x_{t-1}) = x_t$. Then if $x_T = x_{t-1}$, this lets you predict $x_t$. Is this right? How do you ensure that this only applies when $x_T = a$, and not other token types in the vocabulary?
1. Lemma 1: Could you include a proof sketch in the main text? Could you include a longer discussion of the limitations of your simplifying assumptions in Lemma 1 and how they apply to real transformers? The main problem with this section is that I cannot verify that Lemma 1 is true, and that it applies to real transformers in the way you claim that it does. Won't it totally change the dynamics if the other weights are not fixed? I think it's probably a useful result either way, but you should talk about this more.
1. 197: Could you justify this decision a bit more? How do we know this doesn't drastically change the situation from real transformers?
1. 308: Can you include a proof sketch for Lemma 2?

Other questions:

1. Can you clarify which key-value associations you expect the linear layers to learn? Is it the global bigram distribution?
1. 51: If all the examples are generated by a single bigram language model, isn't all of that "global" knowledge? That is, how will you know if the model isn't just memorizing the training data? Are you counting on the fact that the transformer isn't big enough to memorize so many examples? Do you test on out-of-distribution examples that verify that this is the case?
1. 98: What about tied embeddings?
1. 199: I think this assumption is pretty reasonable. But I think it's harder to understand how the induction head would work without a separate query matrix. Could you spend a little time explaining how this would affect the construction?
1. In general it would be very helpful to explain in detail how to design an induction head under these constraints.
1. 208: Again, it would be useful to discuss how W_F would implement this.
1. 219: How?
1. 220: Where do $p_t$ and $p_{t-1}$ come from? Are they sinusoidal positional encodings?
1. Eq 9: Can you explain in words what this is doing?
1. Fig 5: Why is it easier to learn 5 random q than 1 random q? It would be helpful to be reminded that K is the number of triggers per sequence, not the number of token types used as triggers overall.
1. 310: What is $E[x|y = k]$? What is $\hat{p}_W$?

Clarity issues:

1. 43: At this point, I was wondering if you would supplement your empirical analysis of the training dynamics with a mathematical explanation. It would be useful to point out earlier that you will do this.
1. Eq 1: Having $x_t$ on both sides of the equation is confusing. I'm not a big fan of the $:=$ notation.
1. How would the induction head mechanism work in the presence of multiple instances of a in the past context?
1. 120: These sentences seem out of order and are hard to read. It's not clear if $b$ is a variable or just a tag to distinguish $\pi_b$. I don't understand what $\pi_o$ is. I was confused by the difference between $\pi_b$ and $\pi_o$.
1. 123: What is the variable $n$?
1. 124: How is $K$ chosen? Is it randomly sampled or set to a constant value for the experiments?
1. Using i and j as variables for token types instead of input positions is confusing, especially since k is used as an index
1. 126: What is the first token in the sequence? How is that sampled?
1. 126: It would be helpful to provide an intuitive explanation in words of the process that this equation implements.
1. Where is $\pi_u$ used? (I see it is used at 131, but this seems out of order.)
1. 129: What is the tiny Shakespeare dataset?
1. 133: Referring to training details in Section 5 out of order makes this part hard to read. Could you move the training details earlier?
1. 134: Is this on the training data, or on a held-out test set? Do you use a validation set?
1. 134: It would be easier to read this results if they were in a table.
1. Fig 2: The labels for the axes on the first plot are missing. It would be helpful to label the y axes. For the left plot, which version of the dataset was this trained on? Is there a reason you can't show the first layer for both models? I don't understand the significance of the red and green boxes. What are the "previous occurrences of the query"? It would help to highlight the triggers and outputs in the axis labels.
1. 134: At this point I was confused about the significance of using fixed vs. random triggers. But after re-reading 124-126, I understand that for random triggers, the set of triggers is randomly sampled for every sequence, so *every* token type has the potential to be a trigger. The only way they are distinguished as triggers in the training data is that the same bigram appears twice in the same sequence. But this could also happen by chance for non-trigger tokens, right? It would be helpful to add a discussion of this to the main text.
1. 153: What does this notation mean?
1. 159: What does O mean?
1. Eq 5: What does z range over?
1. I didn't realize until Eq 6 that $w_E(z)$ means the $z$th column of $E$. Could you define this notation beforehand?
1. 213: I see that this section answers some of my questions above. It would be useful perhaps to introduce or mention this earlier in some way.
1. 238: Just to clarify, you're using a cross-entropy language modeling loss function, right? I see this is partly answered at 254.
1. 247: In the equation for $W_{*}$, what are $v_j$ and $u_i$?
1. Eq 9: What is $M$? How are the $(i, j)$ picked?
1. 307: This seems to answer one of my earlier questions.

**Limitations:**

As mentioned above, I think the authors should discuss the limitations of their simplifying assumptions about the transformer architecture more.

---

> ### Author Rebuttal · Authors · 2023-08-09
>
> Thank you for the very detailed and encouraging review.
>
> *"The transformer architecture used in this paper is drastically simplified ... the paper requires a much more detailed, readable discussion ..."*
>
> Thanks for raising this point. We hope that our general response to all reviewers provides useful elements to address this concern. In particular, we view simplicity as beneficial to illustrate our model. We also show that various aspects of our model can easily be extended to more complex architectures, although multi-head attention and multiple layers may lead to identifiability issues which make a clean analysis more challenging.
>
> We will do our best to provide a clear discussion of these points in the paper, and will add an appendix with additional experiments and technicalities for extending to more standard architectures.
>
> *"clarity"*
>
> We will do our best to improve clarity and address the points you raise in the questions.
>
> *layer-norm, multi-head, MLPs*
>
> Please see our general response, which addresses these. We will include results on these in the appendix.
>
> *dropout*
>
> We did not consider dropout as it is often not used in modern LLMs (see, e.g., the PaLM paper), but leave its study as an interesting question for future work.
>
>
> *Figure 1*
>
> We apologize for the lack of clarity with this Figure, and will improve this in the updated version. Attn1 and Att2 refer to the desired associative memories for the product $W_Q^\top W_K$ (or just $W_K$ in the simplified model with $W_Q = I$) at each layer.
> You should think of key-query associations as considering two tokens at different positions (instead of separately at each position as you write) and checking if there is a match. E.g. for the first layer, we want $(x_t + p_t)^\top W_Q^\top W_K (x_s + p_s) \approx 1$ for $s = t-1$ and zero otherwise, which is the case for the given choice of associative memory (see also eq.7).
> We will do our best to better represent the full superpositions at each layer, instead of just two elements.
>
>
> *Lemma 1*
>
> We will add a brief proof sketch and add a pointer to the right appendix with the proof.
> The form of the gradient is valid regardless of the current values of the input/output embeddings, so it could potentially apply at any time during training. In particular, if somehow the embeddings stop moving because they converged to a good place, then the weight matrix will end up learning an associative memory with these new embeddings. In practice (see above) we saw that associations are still correct when training embeddings. This could mean the embeddings converge quickly to something stable, and the weight matrices use those final directions to store associations.
>
> *197 (why freeze layers)*
>
> See our general response. Basically our motivation was to get the simplest possible model for which we know what the solution should look like ("identifiable"), in order to have a clean and concise description of training dynamics, both empirically and theoretically. Also, if a good "lazy" solution exists where some of the weights don't need to move, it will likely be an easier solution for GD to find! For attention matrices, this still works in our theory as well, as mentioned in the general response, and we will include these results in the appendix.
>
> *"..key-value associations you expect the linear layers.."*
>
> Indeed, we expect $W_F$ to learn the global bigrams (see eq.8 and the comments below it), at least initially. This would be the case approximately if the two attention layers were not present, at least for non-trigger tokens (following Lemma 1). Our hypothesis for why this also holds initially with attention layers is that all elements in the input superposition to $W_F$ that are not the current token will be mostly noise, since all previous tokens are independent of the next token given the current token (assuming we're on a non-trigger token), so they will get filtered out in the update with enough data (as in Lemma 2).
>
> *"global knowledge"*
>
> While it is possible that a transformer with very large MLP layers could memorize all sequences in order to guess the correct output token after a trigger, this would likely be intractable since the number of such possible sequences grows exponentially with sequence length. Yet our model gets near-perfect prediction accuracy on tokens even near the end of the sequence, suggesting it uses a better approach. The empirical study also shows that the model is making use of the induction mechanism for predictions. We also tried replacing output tokens at test time, and saw correct predictions (though this generally only works for outputs seen during training, in fact if a token isn't used as an output, we see that the corresponding outer product simply isn't stored in $W_O^2$, see our response to uvp3 for more on this).
>
> *other questions*
> * tied embeddings: these also work in our formalism.
> * L199: it is only the product $W_Q^\top W_K$ that affects predictions, which is why we preserve the same expressivity by fixing one to the identity.
> * 208: this is shown in eq. (8)
> * L219 "how?": thanks to near-orthogonality, any embedding in the superposition that does not appear in the outer products gets filtered out
> * L220: $p_t = w_P(t)$ by definition, and is fixed at random initialization
> * eq 9, fig 5: we will clarify these. $K$ is indeed the total number of tokens used as triggers (see L124). Larger $K$ makes it easier to learn the induction head because there is a lot more data involving trigger-output pairs. The values of K used in experiments, if not shown, are given in Appendix D.
> * L310: we will clarify ($E[x|y=k]$ is the conditional expectation of $x$ given $y = k$, and $\hat p_W$ are softmax model predictions for parameter $W$ respectively)
>
> Thank you for the many other comments and suggestions regarding clarity, we will happily incorporate them and will do our best to improve clarity in the updated version of the paper, if it is accepted.

---

> > ### Comment · Reviewer_PfRr · 2023-08-11
> > **Response**
> >
> > Thank you for your response.
> >
> > Weakness 1, MIQ1, MIQ4, OQ3, OQ4: I do agree that simplicity is beneficial for analysis, but the simplified version needs to be tethered to unsimplified transformers in some way, or else the findings of the paper are no longer relevant. I still believe the paper needs at minimum a longer, focused discussion of each of these limitations and how the simplified and unsimplified versions connect. The additional empirical results you have mentioned in the rebuttal will be helpful in arguing this.
> >
> > MIQ2: How close was my original interpretation? And there is no limit to the number of vectors that can be in superposition, not 2 as suggested in the figure, right?
> >
> > MIQ3: Some experimental results demonstrating that the embeddings first converge quickly to an optimum would be helpful here, and in addressing Weakness 1.
> >
> > OQ1:
> > > While it is possible that a transformer with very large MLP layers could memorize all sequences in order to guess the correct output token after a trigger, this would likely be intractable since the number of such possible sequences grows exponentially with sequence length.
> >
> > This is a good point and worth mentioning in the paper.
> >
> > OQ3: Can you explain this in more detail?
> >
> > OQ6: Thanks. This ties into Weakness 2.
> >
> > OQ7: Thanks, this is worth mentioning in the paper.
> >
> > OQ8: Ok, somehow I wasn't able to find this definition in the paper.

---

> > > ### Author Response · Authors · 2023-08-15
> > >
> > > Thank you for your comments.
> > >
> > > Regarding the links between our simplified architecture and the more standard transformer, we agree that the main paper requires more discussion (in addition to the new results that we'll including in the appendix), and we are planning to add a discussion at the end of Section 4.
> > >
> > > **MIQ2**
> > >
> > > What you described in your initial review is indeed a good interpretation of what's going on.
> > > What you write as $\text{query}(x_t, t) = t-1$ can be translated into our associative memory viewpoint as the following constraint on $W_Q$: $W_Q(x_t + p_t) \approx p_{t-1}$. This naturally leads to the desired associations when $W_K$ is the identity matrix: $(x_t + p_t)^\top W_Q^\top W_K (x_s + p_s) \approx p_{t-1}^\top W_K (x_s + p_s) \approx p_{t-1}^\top p_s$, which is non-negligible and close to one only when $s = t - 1$, as desired.
> > > A small difference with our model in Eq.7 is that we use $\text{key}(x_t, t) = t+1$ and $\text{query}(x_t, t) = t$, i.e., the roles of key and query are swapped compared to what you wrote, but this leads to the same associations.
> > >
> > > One thing worth mentioning that seems to be missing in your description is the "remapping" done by the first value layer: it is important to use $\text{value}(x_t, t) = w_1(x_t)$ (using notations from Figure 1) instead of just $x_t$, where $w_1(x) = W_O W_V x$ essentially remaps the embedding $x$ to a new, orthogonal embedding. This ensures that the second layer attention head matches $x_T$ with tokens $x_t$ whose *previous remapped token* $w_1(x_{t-1})$ matches with $x_T$, while without this remapping, it could just as well match tokens that are themselves the same as $x_T$.
> > >
> > > Regarding the question about ensuring the mechanism only applies to certain tokens: when only certain tokens act as triggers (as in the "fixed trigger" setup), this can be achieved by only storing associations for the relevant tokens in the second key/query memories (see the expression for $W_K^2$ in Eq. 7: the summation is only over $k \in Q$, i.e., the set of triggers). For the case of "random triggers", as we discuss at the end of Section 4 (L233-237), the induction head may be active for all tokens, and the model seems to sometimes prefer using it over the global bigram model, particularly if its output is not a frequent next token according to global bigrams (this should indicate that the current token may be more likely to be a trigger in the current sentence).
> > >
> > > Finally, you are correct that there can be many more vectors in superposition. If the attention heads are sparse and only select one token, then we may expect the superpositions after layer $\ell$ to have $O(\ell)$ elements (O() is the [big O notation](https://en.wikipedia.org/wiki/Big_O_notation) which we use throughout the paper). If the attention is more spread out, we may expect more elements, for instance at initialization, the attention is near-uniform, so that we would have the average of all token embeddings in the sequence already at the second layer.
> > >
> > > **MIQ3**
> > >
> > > Thanks for the suggestion, we will include plots of gradient norms, which seem to decrease quickly for the embed/unembed layers, indicating that these layers do not move much later in training.
> > >
> > > **OQ3**
> > >
> > > The associative memories we consider rely on two sets of near-orthonormal embeddings $(u_i)$ and $(v_j)$, where the former are inputs and the latter outputs of some matrix (see Eq. 3). Nothing stops these two sets of embeddings to be the same, and setting $w_U(k) = w_E(k)$ for all $k$ provides an example of this in the context of Lemma 1. This is precisely the case of tied embeddings.
> > >
> > > In practice we observed that using tied embeddings tends to slow down training in our setup, possibly because of the additional correlations it induces, and more importantly because it reduces the overall number of near-orthogonal directions by identifying input and output embeddings, e.g., the output of $W_O^2$ according to eq. (7) would now be an input embedding that may be confused with the embedding of the current token in the residual stream. Nevertheless, it is possible that this weight sharing is beneficial in some tasks for larger models.
> > >
> > > **OQ 1, 6, 7, 8**
> > >
> > > We will clarify these. For positional embeddings, these are defined in Eq. (1), and we will consider dropping the overloaded notation $w_P(t)$, which isn't currently used much, and sticking to the $p_t$ notation throughout the paper.

---

> > > > ### Comment · Reviewer_PfRr · 2023-08-16
> > > > **Response**
> > > >
> > > > Great, thank you for the thorough explanations.
> > > >
> > > > MIQ2, OQ3: Thanks, this is clearer to me now, and I appreciate the addition of details I missed. I think similarly explaining these details in the paper would go a long way toward improving the clarity of Section 2 and Fig 1.
> > > >
> > > > At this point I think my most pressing questions have been addressed, so I am raising my score.

---

> > > > > ### Author Response · Authors · 2023-08-17
> > > > >
> > > > > Thank you for this message. We are glad to hear that our clarifications helped, and will do our best to include them in the updated paper. We appreciate the increase in score!

---

### Official Review · Reviewer_hHJQ · 2023-07-07

**Soundness:** 3 good
**Presentation:** 3 good
**Contribution:** 3 good
**Rating:** 8
**Confidence:** 3

**Summary:**

Given the blackbox that large language models are, this paper tries to use a small simplified view of a 2 layer transformer model, and a synthetic task to understand how global and in-context language statistics are learned by the transformer model.

By freezing specific layers, The authors show how the memory recall and in-context accuracy varies.

**Strengths:**

The paper presents a nice setup with the simplified transformer model, and the synthetic task for analysis of transformer architecture. The current analysis are great, but just the start and the foundation laid could be useful for future explorations.

The proposal around how the global memory and the in-context induction work is supported (partially) through the experiments by freezing various layers.

With the experiments, they are able to show that the global bigram statistics are learned faster and the induction heads for in-context bigrams are developed in subsequent training steps. Additionally, the conclusion that having diverse training distribution leads to better generalization is great. So this is a good started to understanding the transformer LLM blackboxes, and the results may apply on not just text, but other modalities as well.

**Weaknesses:**

- The results are still very preliminary. The transformer model is simplified, and so is the dataset. It is unclear whether a larger transformer model would use a similar paradigm for learning. Essentially, it might be possible that the results do not extend to larger models and additional experiment needs to be done to verify that.

- Transformer models have a lot of other components, and it would be nice to study how they impact the memory of these models.

**Questions:**

- In the multi-head attention, was any experiment perfomed to understand whether for a fixed model dimension, is it better to have larger number of heads or smaller number and how that affects memorization?

- Was there any experiment performed where we just have attention (no feed forward layer), and how that affects global memorization?

**Limitations:**

Discussions could mention that this analysis may not extend to larger versions of the model, or provide empirical proof that it does.

---

> ### Author Rebuttal · Authors · 2023-08-09
>
> Thank you for the helpful review and for the positive assessment of our work.
>
> *“The results are still very preliminary…” "Transformer models have a lot of other components..."*
>
> This is the main topic of our general response post, which we hope provides a valid justification of our approach, as well as as new results in this direction.
>
> In summary, we chose a simple setup in order to provide a more accessible and intuitive description of the weight matrices and their learning dynamics. That said, the “memory” viewpoint for the weights, as well as the insights on training dynamics, extend naturally to more complex architectures, since they are basically a consequence of the fact that inputs and outputs of a weight matrix are embeddings. We will include new experiments and theory justifying this, as presented in the general response. The main difficulty with more complex tasks and architectures is that different mechanisms can be implemented in different layers/heads, making them harder to identify and track. Tackling this in well-chosen or general problems is an important future direction.
>
>
> *"In the multi-head attention..."*
>
> We did not experiment extensively with multi-head attention, since a single head at each layer is sufficient for the induction head mechanism needed for our task. Nevertheless, we ran basic experiments using multiple heads on our task, and observed that the associations made by the second-layer attention tends to distribute across different heads (i.e. different tokens are handled by different heads), while the previous-token mechanism at the first layer seemed to use only one of the heads.
>
> *"Was there any experiment performed where we just have attention..."*
>
> Thanks for the suggestion. We tried an attention-only model, and noticed that the loss on global tokens decreases more slowly than with the feed-forward layer, and ends up at a higher loss number (but note that training embedding layers could circumvent this). For fixed triggers, the task becomes a bit easier and looking at attention maps for the second layer shows that the attention usually focuses on the current token, except for the fixed triggers which need the induction mechanism.
>
> Thanks again for your encouraging review. Please let us know if you have any other questions of concerns.

---

> ### Comment · Reviewer_hHJQ · 2023-08-15
>
> Thanks for the clarifications! I have read through the response and will keep the original scores.

---

> > ### Author Response · Authors · 2023-08-17
> >
> > Thank you for keeping your high score of 8!

---

### Official Review · Reviewer_uvp3 · 2023-07-19

**Soundness:** 4 excellent
**Presentation:** 4 excellent
**Contribution:** 2 fair
**Rating:** 7
**Confidence:** 3

**Summary:**

This paper studies the dynamics of how induction heads emerge in LLMs during training. The authors describe a simple synthetic task to test their hypotheses on, outline a plausible implementation of an induction head based on associative memory, and describe both empirical and theoretical observations.

**Strengths:**

* The paper is well-written and generally easy to follow. Motivations, conclusions, and analysis are crystal clear.
* The authors study an interesting question: what mechanism do LLMs implement induction heads with, and how might it emerge during training?
* The methodology is clean and nice! I like the inclusion of "memory recall probes," as they are a direct measure of what you're claiming. The results are also appropriately framed in the context of the merits & limitations of the experimental setup.

**Weaknesses:**

* Can you better explain the significance of the results? My interpretation is that you framed the $W$ matrices as associative memories and found an order in which they seem to be learned. I'd love to hear what you think the "so what" of this is! What interesting things can we do with this understanding? What more can we learn? Can we make the induction heads better? Implement them manually? Do you expect the learnings here to help us understand larger, more complex models?
* It's sort of implied that the framing of induction heads as associative memory is new. What are the other competing mental frameworks? How does this one compare?
* It'd be interesting to see experiments that vary the dimensionality of the vectors/matrices. In my experience, superposition is rampant, and LLMs often try to stuff a lot into a not-so-large $d$. How do you expect the findings to change when you can't assume near-orthonormality? Other than a degradation in performance?

Minor things:
* Figure 1 is nice! It'd suggest labeling the words in the caption with colors or letters, so it's immediately clear what refers to what.

**Questions:**

* Did you check that the actual weights learned by your toy model match the solution you made in any meaningful way? Maybe exactly, modulo a simple transformation or something? Did you find any interesting surprises?
* Did you have a chance to test any hypotheses about superposition experimentally, even preliminary results? If so, what did you find? Any unexpected things? Negative results ok! I'd imagine it's quite relevant, and is a better representation of what really happens in transformers.

---

> ### Author Rebuttal · Authors · 2023-08-09
>
> Thank you for the encouraging review and helpful suggestions. We are glad you found our work interesting, well written, and that you liked our methodology!
>
> *“Can you better explain the significance of the results? …”*
>
> Thanks for asking this, it is definitely something we should have discussed more in the submission -- we will comment on it more in the next version. At its core, we hope our work can provide a new language to reason about the internals of transformers, and how they are affected by learning dynamics. Here are examples of areas where it could be useful:
> * architecture/algorithms: improve optimization algorithms and architectures/initialization by studying how they impact the form of the learned memories
> * choice of pre-training data: induction heads can be learned more quickly when [trigger, output] pairs are more frequent and more diverse -- perhaps this could extend to more general “reasoning” problems: does synthetic and diverse logic data help? is this why training on code seems useful?
> * interpretability: presumably, knowing the structure of weight matrices can allow a more fine-grained understanding of what each transformer block is doing
> * fine-tuning vs model editing: if gradient updates are just adding/reweighting outer products to our memories, can we do fine-tuning in a more controlled and targeted way by manually changing appropriate weights?
>
> *“Do you expect the learnings here to help us understand larger, more complex models?”*
>
> Yes! The nice thing about this model is that it just assumes that the inputs and outputs of a weight matrix are embeddings (or superpositions thereof), which is basically true anywhere in a transformer other than for embedding/unembedding layers. The main difficulty is redundancy: many different parts of a large model may implement similar mechanisms, so that they may become harder to pin-point and identify during training. This is what motivated us to simplify the architecture into an identifiable model, and similar modifications may work on more complex tasks, but we expect that more general problems will require more work on the interpretability/identification side of things.
>
> See our general response for more discussion on this, and additional results in this direction.
>
> *“It's sort of implied that the framing of induction heads as associative memory is new…”*
>
> Thanks for the question. A perhaps more natural model, which we were initially inclined to believe when we started this work, would be that weight matrices found by training were a bit more “idealized”: for instance, copying could be done with an identity value matrix at the second layer. While recovering matrices like the identity could happen in some settings, we found that in general it is difficult to obtain. One thing we tried in order to see why this intuition fails is to use two different sets of output tokens at training vs test time. Even with tied input/output embeddings (so that the identity approach might actually work on the test tokens), the accuracy on test output tokens is much worse than on train tokens, particularly when the dimension is large [in low dimension, the outer products could span the entire space and actually give something close to the identity, but the overall accuracies are then lower since the embeddings are “less” orthogonal]. We’d be happy to include this discussion in the appendix.
>
> *“…experiments that vary the dimensionality of the vectors/matrices…”*
>
> Figure 7 in the current appendix shows some basic results in this direction, and we’ll include more experiments that study the effect of d and sample size for the one-gradient-step scenario. In practice, we see that higher d definitely helps store things “more quickly” (in terms of iterations and amount of data), but even d=64 is sufficient for our setup.
>
> Regarding superposition, you’re right that there are typically lots of things in the residual stream, but we would argue that d is quite large in LLMs (at least in the thousands), and you can have lots of near-orthogonal embeddings (in fact, exponential in d). The dimension of attention heads is usually much smaller, but we expect that the Q/K matrices filter out a lot of irrelevant embeddings from the residual streams (using a similar process to what we describe in sec. 6), so that the low-d vectors contain just one or a handful of directions.
>
> *“Figure 1 is nice! It'd suggest labeling the words”*
>
> Thanks for the suggestion! We will change this in the final version if the paper is accepted.
>
> *“Did you check that the actual weights…”*
>
> Based on your suggestion, we looked a more granular maps illustrating $(v_j^\top M u_i)_{ij}$ for a desired memory M, instead of just the recall metrics. These do indeed show that the desired associations have much larger values than the remaining items, though we also notice that different input tokens may lead to different magnitudes, especially for $W_K^2$, reflecting different frequencies of triggers in the data.
>
> *“Did you have a chance to test any hypotheses about superposition..”*
>
> Thanks for the suggestion. In our controlled setup where embeddings are just nearly-orthonormal random vectors (including "remapped" random vectors), it is quite easy to check which embeddings are present in a given representation/superposition, by just taking the inner product with each such embedding. Of course this detection becomes more difficult/noisy when there are many elements (e.g. for the initial average attention in a long sequence), but we found that it works reliably for a handful of elements in large enough dimension. We'd be happy to include this in the appendix.
>
> Again, many thanks for your comments and suggestions. We hope these clarifications and improvements may help increase your score. Please let us know if you have any more questions or concerns.

---

> > ### Author Response · Authors · 2023-08-18
> >
> > Dear reviewer uvp3, thanks again for the helpful comments and suggestions. As the discussion period nears its end, please do let us know if you have any additional questions or concerns with the paper. We'd be happy to address them. --the authors

---

> > > ### Comment · Area_Chair_JYDs · 2023-08-19
> > > **Thanks for your response!**
> > >
> > > Dear authors,
> > >
> > > Many thanks for your submission and detailed response!
> > >
> > > We have pinged reviewer uvp3 to acknowledge your response and raise follow-up questions, if they have any.
> > >
> > > Best regards,
> > >
> > > Your AC

---

> > > ### Comment · Reviewer_uvp3 · 2023-08-19
> > >
> > > A big thank you to the authors for your efforts during rebuttal — I've gone through your response, the general rebuttal, as well as comments on the other reviewers' posts. It's clear that a lot of care went into them.
> > >
> > > I think this paper makes a solid, technically-sound contribution to our (nascent) understanding of how induction head mechanisms are learned. The authors have addressed my primary concerns around framing, so I will increase my score.
> > >
> > > FYI, though, I'm still a little hazy about what the immediate next steps of future work are. On "interpretability," I don't quite see the direct connection between this work and being able to understand exactly what the weights do; that sounds like a far more complicated problem. On "model editing," how does this work tell us which weights to edit? On "pre-training data choices," could you give me a specific example of what you'd like to try, something that sounds practical?
> > >
> > > I would love to be convinced otherwise regarding these points! Even if speculation, a more *concrete* set of things you think are promising to study (with specific connections to learnings from this paper) would be helpful to me, and I'd be happy to increase my score further :)

---

> > > > ### Author Response · Authors · 2023-08-21
> > > >
> > > > Thank you for your reply, and for increasing your score! We highly appreciate it.
> > > >
> > > > Without going in too much detail (we'd love to talk more once this is all over..!), here are some more concrete directions:
> > > > * Interpretability: our work suggests that you could find/verify input-output associations of a given matrix by testing it against pairs of embeddings or remapped embeddings from different layers. This can avoid the need for "[mean ablation](https://arxiv.org/abs/2211.00593)" since you'd only have "pure" embeddings instead of superpositions, though in practice the large number of possible remappings means there may still be a need for circuit identification (e.g. via some causal analysis).
> > > > * Editing: adding/subtracting outer products has already been done in the context of factual recall in MLPs (see [here](https://rome.baulab.info/)) -- our work leads to a better understanding of this procedure by relating it to associative memories, and suggests it can be done elsewhere as well, e.g., in either Q/K or V/O mechanisms in attention.
> > > > * Data: our view here is that the memory viewpoint can help reason about what data can help to learn certain mechanisms/memories more efficiently, as illustrated in Figure 4. For instance, in the context of a code model, filtering for documents with more "logic" than verbosity (e.g. large fraction of parentheses/keywords/indents per token) may result in faster learning of some reasoning heads (similar to using more triggers in Figure 4), versus global memorization. The middle plot in Figure 4 also suggests choosing more diverse and uniform distributions for the output tokens for better OOD performance -- in the context of code, this could suggest synthesizing more diverse data by randomizing variable names.
> > > > * As a side note, our analysis also provides insights on large width scalings for learning rate and initialization, and sheds a different light on maximal updates for feature learning when using input embeddings initialized with variance $O(1/d)$ instead of $O(1)$ (as in [muP](https://arxiv.org/abs/2203.03466)).

---

### Official Review · Reviewer_hu6p · 2023-07-26

**Soundness:** 4 excellent
**Presentation:** 3 good
**Contribution:** 3 good
**Rating:** 6
**Confidence:** 4

**Summary:**

Authors perform an in-depth study of the toy case of learning associate recall task using causal Transformers with the aim to understand the emergence of in-context learning abilities during training. Informally, they propose a modified bigram distribution where after sampling a sequence, for a special set $Q$ of pre-determined "trigger" tokens, at every occurance of a token $q$ from $Q$, they replace its following token with the token that followed $q$ at its first occurrence in the sequence. I.e. $[...q\ r ....q\ s] \mapsto [...q\ r ....q\ r]$. Hence, for $q$, the model is required to determine $r$ first from the context and, at every following occurance of $q$ output $r$. The authors prove that a simplified 2-layer Transformer (the only non-linearity is softmax) can learn this task via gradient descent and verify this empirically.

**Strengths:**

1. The problem is well-motivated as large language models demonstrate impressive in-context learning abilities and hence it is beneficial to understand how this ability develops.

2. The theoretical statements are non-trivial, interesting and not straightforward to prove.


**Weaknesses:**

1. The associative recall task has been studied in the context of in-context learning of attention-based models before and its not fully clear what novel contributions the authors have made.

2. Given the simplicity of the toy task/model the practical implications and the generality of the findings are unclear (other tasks / architectures).

**Questions:**

.

**Limitations:**

There is no discussion of the limitations.

---

> ### Author Rebuttal · Authors · 2023-08-09
>
> Thank you for the thoughtful review.
>
> *“The associative recall task has been studied… novel contributions”.*
>
> Indeed, several works have looked at similar tasks. Nevertheless, to our knowledge we are the first to have a precise picture of (pre-)training dynamics and as a result, a precise understanding of the form of the weights, in a way that extends to multiple layers. The structure of our bigram data model we introduce is also simple enough that it is amenable to theoretical analysis. Overall, we believe that our study provides many new insights about the internal structure of transformers during pre-training, and can pave the way for new improvements, e.g. for optimization algorithms, data selection, fine-tuning, model editing, and interpretability.
>
> *“…practical implications and the generality of the findings are unclear...”*
>
> Indeed, this work is a first step. That said, our associative memory viewpoint for weights naturally extends to more complex architectures since gradients will take similar forms, and we chose a simple task precisely to make the viewpoint more transparent and understandable. We also simplified the architecture in order to make the role of each component identifiable, while for a more complex model, there may be many different solutions (e.g. the previous token head could happen in many different layers and attention heads).
>
> Other tasks are definitely interesting but might involve very different mechanisms which we believe will require separate studies, yet, copying mechanisms like ours are likely crucial in many different tasks (as the induction head papers illustrate), and enable more complex reasoning operations, e.g. with a semantic hierarchy that may be learned in multiple layers.
>
> We expand on this point much more in the general response to all reviewers, and there we discuss additional experiments/theory that shows our viewpoint extends to more complex settings.
>
> Please let us know if this response helps with your assessment of the paper, we’d be happy to answer any additional questions.

---

> > ### Comment · Reviewer_hu6p · 2023-08-10
> > **response 1 to rebuttal**
> >
> > Thank you for addressing some of my concerns - after going through your responses to my review (and to other reviews) I am increasing the score.

---

> > > ### Author Response · Authors · 2023-08-15
> > > **Thank you!**
> > >
> > > Thank you for the increase in score! We appreciate this.

---

### Author Rebuttal · Authors · 2023-08-09

We would like to thank all reviewers for their insightful feedback and valuable comments.
We are happy that most reviewers found our work relevant and significant. Indeed, we believe that our insights on the internals of transformers can pave the way for improved methods in several aspects of LLMs, including better optimization algorithms, data selection, fine-tuning, model editing, and interpretability.

We provide a response below to a shared concern regarding our simplified setup, and respond to individual reviewers in separate replies.

**Why a simplified architecture**

The simplified architecture compared to common transformers/LLMs was a concern for multiple reviews.

Our goal was to simplify the model as much as possible to ease the understanding of what is happening, while ensuring that the model is still rich enough to capture the relevant phenomena to solve the task. It would definitely be much more cumbersome to illustrate the memory viewpoint, and theoretically study training dynamics on a model where many more components are trained. We hope the simplicity of our architecture can help provide better intuition for what we believe to be a key internal mechanism in all transformer models.


**More components**

In line with the reviewers' suggestions, we ran additional experiments to check if we obtain similar mechanisms with more trained components.
In particular, we trained a similar two-layer model with the following modifications:
* train all parameters (including input/output embeddings and all four attention matrices at both layers)
* use a ReLU MLP feed-forward layer at the second layer (instead of linear)
* add pre-normalization layers

We found that despite this added complexity, the "memory probes" described in Section 5 still display the same associations empirically. Concretely, we replace $W_K^\ell$ by $W_Q^{\ell\top} W_K^\ell$ in eq.(7), and~$W_F x$ by $MLP(x)$ for the feed-forward probe (note that these can still be defined even if the parameters are changing). The three recall probes still converge to 1, while the KL probe on the feed-forward layer decreases.

This shows that this more realistic model still identifies the same mechanisms empirically.
On the theory side, we can also easily show that gradient steps on each attention matrix from random initialization still recovers similar associations despite the redundancies (in the pairs Q/K and V/O). Additionally, we can show that layer-norm and MLPs preserve the associative memory form of the weights:
* Adding layer-norm essentially adds a projection operation to the rank-one terms in the gradients. The projection essentially drops outer product terms for which the association was already present. In particular, it plays no significant role at random initialization, thus does not change our single-gradient-step analyses, but is likely important to study optimization stability.
* For the MLPs weights, the outer product terms in gradients involve non-linear mappings of superpositions, which may more easily capture interactions between different elements of the superpositions.


We'd be happy to include these additional experiments and technical details in the appendix, if the paper is accepted.

**More heads and layers**

While the experiments above show some robustness to common complexities, the single-head, two-layer architecture remains very simplified compared to large models.
Unfortunately, understanding more complex architectures with multiple heads and more layers is more challenging since there is a lot of redundancy and it is unclear which layer or head may be implementing different mechanisms.

For instance, we tried using 4 heads instead of one in the two-layer model, and the second layer induction mechanism appears to distribute across all the heads, each head taking care of different tokens (though interestingly the first layer previous-token mechanism seems to only use a single head). This is in contrast to the single-head induction head mechanism we describe, which becomes identifiable (i.e. we know what each matrix ends up doing) thanks to the reduced redundancies.

Extending our work to general models and tasks will thus require more work--in the spirit of interpretability--to first identify which parts of the model are implementing different mechanisms, before understanding dynamics. Nonetheless, our results suggest that within each block, gradient dynamics naturally lead to weight matrices that may be interpreted as associative memories.

We hope this post clarifies our motivations for the simple model, and we will do our best to make this discussion clearer in the paper.

---

### Decision · Program_Chairs · 2023-09-21

**Decision:**

Accept (spotlight)

**Comment:**

This work conducts a careful mechanistic analysis of how in-context learning capabilities emerge during training of a simplified version of the transformer architecture, on a toy task. The authors focus on two complementary mechanisms of (a) in-context learning through induction-heads, and (b) an "associative memory" in the attention weight matrices that captures the global data distribution. They give both mathematical and experimental justifications for the emergence of these mechanisms, and show concrete connections between learnability and attributes of the training data distribution.

The main criticism raised by the reviewers was about the simplicity of the analyzed architecture compared to standard language models (LMs) and of the evaluation setup compared to more challenging natural tasks. These concerns were addressed during the rebuttal, as the authors extended their experiments to consider more components from the standard transformer architecture, and explained the non-trivial effort required to go beyond this, which seems reasonable to leave for future work.

Overall, this is a rigorous study that provides valuable insights into fundamental questions of how transformer-based LMs obtain ICL capabilities during training and balance between ICL and generation from parametric knowledge.